# Observed and Predicted Trends in Icelandic Snow Conditions for the period 1930-2100

Darri Eythorsson[a], Sigurdur M. Gardarsson[a], Andri Gunnarsson[b] and Oli Gretar Blondal Sveinsson[b]

[a] Faculty of Civil and Environmental Engineering, University of Iceland, Iceland
5 [b] Research and Development Division, Landsvirkjun, Iceland

*Correspondence to*: Darri Eythorsson (dae5@hi.is)

**Abstract.** This study presents an estimate of historical snow conditions in Iceland and a projection of these conditions, given different emission scenarios. Historical snow conditions were estimated using in situ observations from manned meteorological stations over the period 1930-2021 and by remotely sensed observations from the MODIS instruments over the period 2001-10 2021. Historical and future climate conditions, as described by each of the 21 General Circulation Models (GCM's) from the 5th iteration of the Coupled Model Intercomparison Project (CMIP5) as contained in the NASA Earth Exchange (NEX) Global Daily Downscaled Projections (GDDP) dataset, were used to simulate snow conditions in Iceland over the period 1950-2100 under the Representative Concentration Pathways (RCP) RCP4.5 and RCP8.5 with the SNOW-17 model. The results show an increase in the average annual Snow Cover Frequency (SCF) over the historical record detected both in the in-situ (1930-2021) 15 and remotely sensed data (2001-2021). Average annual snow depth measurements also revealed an increasing trend over the historical record. Simulated snow conditions show a substantial decrease in both Snow Water Equivalent (SWE) and SCF over the period 1950-2100, a trend more pronounced under RCP8.5 as compared to RCP4.5.

## 1. Introduction

Icelandic climate is categorized as maritime, with mild winters, cold summers, strong winds, frequent precipitation and large 20 spatio-temporal variations in weather and micro-climate (Bjornsson et al., 2007; Ólafsson et al., 2007). It is strongly influenced by ocean conditions in the Northern Atlantic (e.g. Massé et al., 2008) and mass balance trends of Icelandic glaciers are highly correlated with changes in large-scale ocean circulations (Eythorsson et al., 2018). Since the last glacial maximum the average annual air temperature in Iceland has increased about 4°C (Geirsdóttir et al., 2013; Knudsen et al., 2008; Langdon et al., 2011; Larsen et al., 2011; Sicre et al., 2011). The average air temperature in Iceland has risen by 0.08°C/decade since the 1850's, 25 comparable to the global average, and by 0.5 °C/decade over the period 1980-2016 (Bjornsson et al., 2018). Since 1890 the Icelandic glaciers have lost about 16% of their mass and 18% of their surface area, contributing about 1.5 mm of global sea level rise (Aðalgeirsdóttir et al., 2020; Bjornsson et al., 2013) and are expected to lose most of their remaining mass over the next two centuries at current pace (Adalgeirsdottir et al., 2006; Bjornsson & Palsson, 2008; Jóhannesson et al., 2004; Schmidt et al., 2020). Runoff in Iceland is generally expected to increase in winter as less water is stored in snowpack and runoff from

glaciers is expected to increase until at least the middle of the 21$^{st}$ century (Blöschl et al., 2017; Jónsdóttir, 2008), the rate of which is expected to vary depending on ocean conditions in the North Atlantic, where recent cooling has led to a slowdown in mass loss of Icelandic glaciers (Noël et al., 2021). Spring melt is generally predicted to begin earlier and autumn snow cover to occur later (Johannesson et al., 2007). Analysis of a recently developed gap filled MODIS snow cover product suggests that the snow cover duration has increased during the period 2000-2018 for all months expect October and November (Gunnarsson

et al., 2019). Understanding of future expected changes to snow in Iceland is important for water resources management as it constitutes a considerable portion of the regional hydrological cycle, especially in the interior highlands where the majority of the country's energy production occurs, in hydropower plants fed by glacial rivers.

Snow cover monitoring by satellite remote sensing has been studied since the 1960s and several global snow cover products have been produced based on these observations. (Dong, 2018; Frei et al., 2012; Robinson et al., 1993). The MODIS

instruments on the Terra and Aqua satellites (Dietz et al., 2012) provide a good balance of spatial and temporal resolution, with two daily observations and 500m x 500m pixels (Aalstad et al., 2020). An imporant variable for snow remote sensing is the Snow Cover Frequency (SCF), the number of days with snow cover divided by the number of valid observations per year (Nolin et al., 2021) which is related to e.g. growing season length and habitability (Callaghan et al., 2011). SCF is a key variable in the Earth's energy balance (Cohen, 1994) and can be used to analyze the impacts of climate change on the

cryosphere (Brown & Mote, 2009).

Snow condition estimates by both General Circulation Models (GCM) and Regional Climate Models (RCM) capture the main traits of annual snow cycle but are know to contain biases due to their relatively simple snow schemes (Frei et al., 2018; Matiu & Hanzer, 2022). In general GCM-RCM pair ensembles continuation of the ongoing reduction in average snow conditions until the middle of the 21$^{st}$ century (Verfaillie et al., 2018). Improved estimates of snow conditions have been achieved e.g.

using various re-analysis (e.g. Fiddes et al., 2019), downscaling (Fiddes et al., 2022; Smiatek et al., 2016) and data assimilation methods applied either to GCM-RCM snow projections or projections of snow conditions by different snow models forced with downscaled and/or bias corrected GCM-RCM data (e.g. Hanzer et al., 2018).

Many snow models have been developed and described in the literature (e.g. Krinner et al., 2018; Magnusson et al., 2015). The SNOW-17 model was developed for the US national Water Service where it has been used for operational snow forecasting

for the past several decades (Anderson, 2006). The SNOW-17 model has been applied to several regional climate change studies (Miller et al., 2011; Notaro et al., 2014) and has shown good correlation to MODIS Snow-Covered Area (SCA) observations (Franz & Karsten, 2013) . A key advantage of the SNOW-17 model is that it is a conceptual model which simulates snowpack conditions based on a temperature index which is both more computationally efficient compared to full energy balance models and requires fewer and simpler forcing data variables.

The objective of this study was to analyze observed trends and predict the development of snow conditions in Iceland under different plausible climate scenarios. It presents an analysis of historical and future trends in Icelandic climate and snow conditions. Improved understanding of how local snow resources are likely to respond to changing climate conditions is important as these changes are expected to impact local communities and ecosystems as well as changing the challenges and

opportunities for exploiting natural resources in cold areas (Eliasson et al., 2017). In this study changes to historical snow cover properties were estimated based on both in-situ and remotely sensed observations. Future snow conditions were projected by modelling based on a globally downscaled and bias corrected ensemble of General Circulation Models (GCM) from the 5$^{th}$ iteration of the Coupled Model Intercomparison Project (CMIP5). The novelty of this study is the analysis of an extended dataset of in-situ records of snow conditions in Iceland combined with reliable remotely sensed dataset of snow conditions in the area and the comparison of these observations with snow conditions simulated using a trusted snow model.

## 2. Methods

### 2.1 Tools and Datasets

### 2.1.1 In Situ Snow Observations

Data on in situ snow measurements at manned monitoring stations were acquired from the Icelandic Meteorological Office (IMO) (Icelandic Meterologocial Office, 2021). The data contains all observations and manual measurements of local snow depth (*SND)*, snow cover status (*SNC*), precipitation (*R*), precipitation class (*RTEG*), and a visual estimate of surrounding mountain snow cover status (*SNCM*) for total 266 manned observation stations that have recorded snow data in the period 1930-2021. *SNCM* is measured with the intent to represent snow cover status in the highlands at 550-650 m a.s.l. in the mountains visible from each observation station, as best applicable to the site of each station. *SNC* is measured with the intent to represent the average snow cover status in the near vicinity of the observation station, within 1 km radius (Icelandic Meterologocial Office, 2008).

Figure 1 (left panel) shows the locations of the monitoring stations that have recorded *SNC* continuously for at least 20 years at some time during 1930-2021, the figure shows that the observations are spread around the lowlands near the coastline with more sparse observations in the interior highlands. Figure 1 (right panel) shows the number of IMO stations reporting snow variable observations over the period 1930-2021 and the average annual snow depth across all stations for the same period. The number of stations reporting snow data is below 10 until 1950 and rapidly increases thereafter, the number of stations recording snow cover status increases prior to those recording snow depth, from the 1960's snow depth has been recorded at more than 60 stations. Figure 1 (right panel) shows that the average annual snow depth from all stations has remained similar throughout the study period.

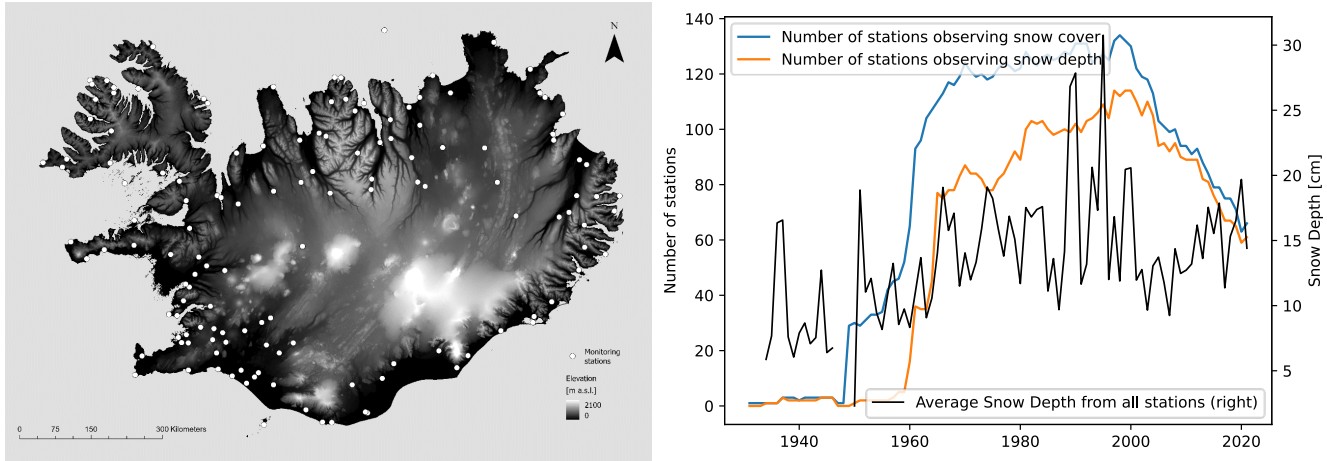


**Figure 1 Left panel: Topography of Iceland** *(National Land Survey of Iceland, 2016)* **and the location of IMO monitoring stations where snow has been measured continuously for at least 20 years in the period 1930-2021. Right panel: Number of IMO stations observing snow variables and the average annual snow depth from all stations over time** *(Icelandic Meterologocial Office, 2021)*.

### 2.1.2 Remote Sensing and Geospatial Data

The MOD10A1.006 and MYD10A1.006 daily snow cover products from the MODIS instruments on NASA's Aqua and Terra satellites (Hall et al., 2016) were used to estimate spatial changes in snow cover over the period of the 2001-2021 water years. The 'NDSI_Snow_Cover' band was used to estimate the presence of snow in each pixel. The band contains the value of the Normalized Difference Snow Index (NDSI) which leverages the fact that snow is highly reflective within the visible spectrum but not the shortwave infrared (Dozier, 1989; Painter et al., 2009). The NDSI values in each pixel are given in a range of 0-

100 where a value of NDSI > 0 often indicates the presence of some snow in the pixel and a value of 100 that the pixel is likely fully snow covered. The NDSI is not to be confused with Fractional Snow Covered Area (FSCA) which is a measure of the fractional snow coverage of a pixel although there exist commonly used transformations between MODIS FSCA and NDSI, including linear (Salomonson & Appel, 2006), inverse linear (Fiddes et al., 2019) and other regression methods (Alonso-González et al., 2021). Results from the Sentinel-2 mission have also shown good performance of other NDSI-FSCA

transformations (Gascoin et al., 2020). Regression methods have shown good performance on medium resolution observations such as from MODIS, whereas higher resolution observations have been shown to benefit from spectral unmixing (Aalstad et al., 2020; Cortés et al., 2014). The 'NDSI_Snow_Cover_Basic_QA' band was used to select observations by quality estimate. A 10 x 10m DEM (IslandDEM) was used for topographical information (National Land Survey of Iceland, 2016).

### 2.1.3 Climate Data

The NASA Earth Exchange (NEX) Global Daily Downscaled Projections (GDDP) dataset (Thrasher et al., 2012; Thrasher et al., 2006) was used as an estimate of historical and future climate. The dataset contains global minimum and maximum near surface air temperatures and surface precipitation rates, as estimated by 21 globally downscaled and bias-corrected CMIP5 GCM's, in 0.2-degree horizontal resolution for the period 1950-2100. It is noted that the CMIP6 version of the NEX-GDDP

dataset was published after the conclusion of the present study and could be considered in future studies (Thrasher et al., 2022).

Daily average temperature was calculated as the mean of daily minimum and maximum temperatures and the ensemble mean was used to represent future climate. Climate and land cover data from the Global Land Data Assimilation System V 2.0 (GLDAS-2) (Rodell et al., 2004) Dataset was used for parameter estimation  and permafrost extent data from the Arctic Permafrost Map (WGS43261) were also used for parameter estimation (Brown et al., 2002).

## 2.2 Data Processing

**2.2.1 In Situ Observations**

*SND* is recorded for all days with snow covered ground, in cm. SNC and SNCM are classified by visual observation as: 0 = no snow, 2 = patchy snow cover, 4 = fully covered ground. (Icelandic Meterologocial Office, 2008). The 1st of April *SND*  was calculated for all stations with more than 20 years of continuous snow depth measurements within the period 1930-2021 (n = 89). The annual Snow Cover Frequency (*SCF*) was calculated for all stations with more than 20 years of continuous snow

cover status observations within the period 1930-2021 (n = 93). *SCF* was calculated as number of days with snow covered ground divided by the number of days in the year, for both fully snow-covered ground (*SNC or SNCM* = 4) and for patchy to full snow cover status (*SNC or SNCM* ≥ 2). SCF was calculated both for observations on the immediate surroundings of the observation site (*SFC*) and on surrounding mountains (*SFCM)*.

**2.2.2 Remotely sensed observations**

Binary snow cover classification was derived from the MOD10A1.006 and MYD10A1.006 snow cover products (Hall et al., 2016). Data from the 'NDSI_Snow_Cover' band was selected for observations with the highest quality estimate ('NDSI_Snow_Cover_Basic_QA' = 0). The daily mean of 'NDSI_Snow_Cover' band was calculated from both snow cover products. Pixels with 'NDSI_Snow_Cover' > 0 were classified as snow cover (1), and other as no snow (0). The average annual SCF was calculated by counting the number of snow-covered days and dividing by the number of days with valid

observations in each pixel, per hydrological year. SCF was calculated based on the highest quality observations, thus excluding lower quality observations as well as missing data due to cloud cover. The availability of MODIS data during polar darkness is a temporal limitation for the data set.

**2.2.3 Snow Modelling**

Daily snowpack in Iceland was simulated for each hydrological year in the period 1950-2100 using the SNOW-17 model

(Anderson, 2006). The model was run in a 0.2-degree resolution with daily average precipitation and temperature data from each of the 21 downscaled and bias corrected CMIP5 GCM's in the NASA NEX GDDP dataset (Thrasher et al., 2012). The model was initialized at the start of each hydrological year in the study period to prevent snow accumulation between years. The model was applied to each of the 21 CMIP5 GCMs in the NASA NEX GDDP dataset and to both the RCP4.5 and RCP8.5

scenarios. These scenarios were chosen to represent both a „business-as-usual" scenario (RCP8.5) and a stabilization scenario
(RCP4.5) where anthropogenic climate forcing is assumed to be stabilized by the end of the century. The SNOW-17 algorithm
was coded in Google Earth Engine (GEE) , the simulations were performed in GEE and the input data were accessed through
the GEE data catalog.

SNOW-17 uses 10 model parameters that must be specified by the user for each location. In this study the SNOW-17
parameters were determined at the model resolution across Iceland based on local topography, ecology and hydrology. The
recommendations provided by the author of the model (Anderson, 2006) were followed for all model parameters except the
melt factors *MFMAX* and *MFMIN* which are key model parameters that describe the relation between surface air temperature
and snowmelt. For the melt factors the methods of Mizukami & Koren, (2008) were followed as they  incorporate information
about the local slope, aspect, shading, vegetation cover and local short wave energy balance. The method for parameter
determination was selected as calibration of parameters influenced by prevailing climate conditions would cause bias across
the timescales considered in this study, although it incurs the uncertainty associated with parameter estimation. The supporting
data used for parameter determination were SWE, surface air temperature, precipitation and snowfall rates, near surface wind
speed, land cover classification and net solar radiation from the Global Land Data Assimilation System V 2.0 Dataset (*GLDAS-2*), a 10x10m Digital Elevation Model (DEM) of Iceland (IslandDEM) and permafrost extent data from the Arctic Permafrost
Map (WGS43261). Table 1 summarizes the description of each of the SNOW-17 model parameters and presents their value
ranges, units, source methodology and supporting datasets used for their determination.

**Table 1 SNOW-17 model parameters and the value ranges, units, source methods and supporting data used for each.**

| Parameter | Range | Units | Supporting data | Methodology |
|---|---|---|---|---|
| Gauge under-Catch Factor (*GCF*) | 1.0 | - | - | Andersson, (2006) |
| Maximum Melt Factor *(MFMAX)* | 0.7 – 2.4 | mm °C$^{-1}$ *6h$^{-1}$ | *IslandDEM GLDAS-2* | Mizukami & Koren, (2008) |
| Minimum Melt Factor *(MFMIN)* | 0.001 – 1.5 | mm °C$^{-1}$ *6h$^{-1}$ | *IslandDEM GLDAS-2* | Mizukami & Koren, (2008) |
| Average wind during rain on snow *(UADJ)* | 0.02 – 0.4 | mm mb$^{-1}$ | *GLDAS-2* | Andersson, (2006) |
| Temperature determining rain/snow *(PXTEMP)* | -1 – 3 | °C | *GLDAS-2* | Andersson, (2006) |
| Base temperature where Melt occurs *(MBASE)* | 0 | °C | - | Andersson, (2006) |
| Maximum Negative Melt Factor *(NMF)* | 0.05 – 0.3 | mm °C$^{-1}$ *6h$^{-1}$ | *GLDAS-2* | Andersson, (2002) |
| Antecedent Temperature Index *(TIPM)* | 0.05 – 0.2 | - | *GLDAS-2* | Andersson, (2002) |
| Physical Liquid Water Holding Capacity *(PLWHC)* | 0.02 – 0.3 | % | *GLDAS-2* | Andersson, (2002) |
| Constant basal Melt rate *(DAYGM)* | 0 – 0.3 | mm day$^{-1}$ | *WGS43261* | Andersson, (2006) |

The yearly April 1st SWE was extracted for each ensemble member. The annual SCF was also calculated for each ensemble member as the number of times a model grid cell contained snow per year divided by the number of days in that year. The April 1st SWE was used as it has a long history of use as a snow metric for streamflow forecasting and the SCF was used as it has been suggested as a more appropriate snow metric for a changing climate (Nolin et al., 2021).

### 2.2.4 Data Analysis

This study analyzed a large amount of data on Icelandic snow conditions with the purpose of studying long term trends in snow conditions based on publicly available authoritative datasets .. The presence of a statistically significant trend in the time series of in situ observed mean annual SCF, SND was estimated using the Mann-Kendall trend test and by using Sens's estimator of slope method for the MODIS observations. Both of these tests have often been applied to trend analysis in snow cover studies (Notarnicola, 2020; Yilmaz et al., 2019). The Sen's slope method was applied to the remotely sensed
observations as it is tolerant to outliers (e.g. Nguyen et al., 2022). The trend test p-values were calculated for the annual SCF and SND timeseries. If $p < 0.05$ the change in the observed data was assumed unlikely to be caused by random variability, indicating a presence of a monotonic trend. The average annual snow rain ratio in Iceland was estimated from ensemble mean of air temperature and surface precipitation data from NEX-GDDP dataset by applying a simple rain/snow partitioning scheme, where precipitation is classified as snow under a set temperature threshold (0°C). Google Earth Engine (GEE) (Gorelick et al.,
2016) was used to access data, perform simulations and analyze results. Statistical analysis was performed using GEE and the SciPy toolbox (Oliphant, 2007). ArcMap 10.7.1 was used to produce maps showing the results.

### 3. Results

### 3.1 Historical Snow Cover Trends

Figure 2a shows the average temperature and precipitation in Iceland over the period 1950-2021 as estimated from the
ensemble mean of the 21 GCMs in the GDDP dataset. The figure shows that both temperature and precipitation have a positive trend during the period while the variability in precipitation has been more than for temperature observations. Figure 2b shows the annual average SCF for all IMO monitoring stations for the period 1950-2021, calculated for local (circles) and mountain (triangles) snow cover status based both on just observations of fully snow-covered ground (SNC or SNCM = 4) and including patchy snow cover status (SNC or SNCM ≥ 2), the in-situ data is shown with a 10-year rolling average and a linear trendline.
The figure shows the average annual SCF estimated from the MODIS Terra/Aqua snow cover products (black markers) for observations above (stars) and below (crosses) 500 m a.s.l. The figure shows an increasing trend for all observations and that the MODIS observations below and above 500 m a.s.l. correspond well with snow cover status observations around the observation sites and in the surrounding mountains, respectively. Figure 2c shows the average annual snow depth (*SND*) of all IMO monitoring stations for the period 1950-2021 with a linear trendline and a 10-year rolling average. Due to the considerable
increase in the number of stations reporting snow measurements in the 1950's, as illustrated in Figure 1, data prior to that was

not used for trend analysis. The figure shows an increasing trend of SND over the period with the highest values recorded by the end of the 20th century.

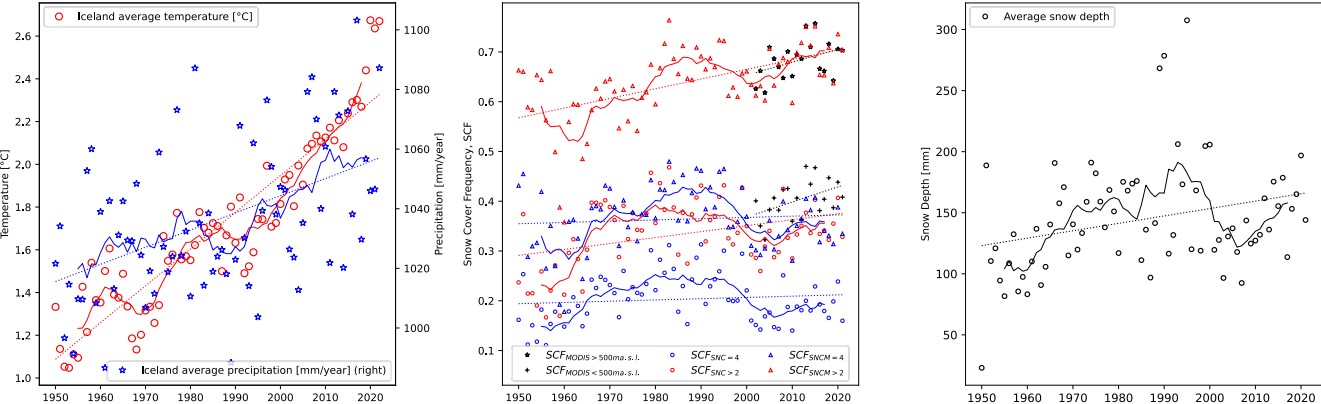

**Figure 2 Left panel (2a): Average annual temperature and precipitation in Iceland over the period 1950-2021 as estimated by the ensemble mean of NASA NEX-GDDP.  Centre panel (2b): Annual average SCF for all IMO monitoring stations for the period 1950-2021 Right panel (2c): average annual snow depth of all IMO monitoring stations. The solid lines represent a 10-year rolling average.**

The results in Figure 2 show that on average both *SND* and SCF in Iceland have a positive trend over the period 1950-2021. The trend is more apparent when considering both fully and patchy snow cover status, (SNC or SNCM $\geq$ 2) and the data reveal considerable natural climate variability. The MODIS estimates of SCF below and above 500 m a.s.l. are comparable to the in-situ estimates of local and mountain SCF, respectively.

Figure 3 shows the estimated average annual Snow/Rain ratio and the projected average annual changes to precipitation and temperature in Iceland over the period 1950-2100 compared to 1950-1960 averages (Temperature = -3.35 °C, Precipitation 1028 mm/year) given two emission scenarios. The figure shows that as both temperature and precipitation are expected to experience a continuation of the ongoing increase from the 1950-1960 average the average annual Snow/Rain ratio across Iceland is expected to decrease continually, from around 0.6 to around 0.2 and 0.1 for RCP4.5 and RCP8.5 respectively. This trend will be apparent sooner at lower elevations where air temperatures are closer to the snow/rain partitioning threshold. At higher elevations the observed increase in precipitation will result in a temporarily thicker snowpack overall, as air

temperatures are further from reaching the threshold, which would offset the increased winter snowmelt and shortening of the
snow cover duration associated with temperature rise until the threshold is reached.

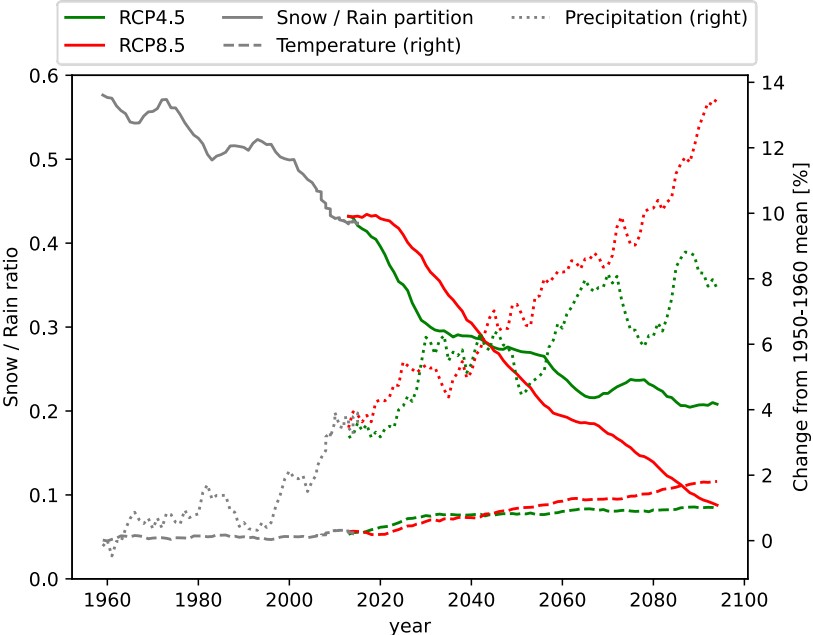

**Figure 3 Estimated Average annual Snow/Rain ratio and projected changes for average annual temperature and precipitation in Iceland for the period 1950-2100 given the RCP4.5 and RCP8.5 emission scenarios.**

Table 2 summarizes the statistical significance of the estimated snow trends, estimated using the Mann-Kendall trend test, for
both the period of extensive historical records (1950-2021) and the MODIS period (2001-2021), in terms of p values. The
values for the historical trends are calculated from 1950 as the number of stations reporting snow data are few prior to the1950s,
as shown in Figure 1.The results show that the increasing SCF and *SND* trends observed in Figure 2 are statistically significant
over the period 1950-2021 for all SCF estimates except for observations of SNCM = 4 (fully snow-covered mountains). Over
the MODIS period 2001-2021 the trend is significant for all metrics except for observations of SNC = 4 and for MODIS
observations above 500 m a.s.l.

**Table 2 Statistical significance of the linear SCD and SCF trendlines, estimated using the Mann-Kendall (MK) trend test, for the full historical period (1950-2021) and the MODIS period (2001-2021). Statistically significant trendlines at the α = 0.05 level are shown in bold.**

|  | Trend [% per decade] | | p-value | |
|---|---|---|---|---|
|  | 1950-2021 | 2001-2021 | 1950-2021 | 2001-2021 |
| SCFM (SNCM ≥ 2) | 0.1 | 4.3 | **1.4 x 10⁻⁶** | **0.02** |
| SCFM (SNCM = 4) | 0.0025 | 2.1 | 0.9 | 0.07 |
| SCF (SNC ≥ 2) | 0.1 | 3.7 | **0.025** | **0.01** |
| SCF (SNC = 4) | 0.002 | 1.9 | 0.6 | 0.06 |
| *SND* | 0.05 | 3.0 | **0.006** | **0.002** |

| | | | | | |
|---|---|---|---|---|---|
| MODIS below 500 m a.s.l. | - | 2.9 | | - | **0.04** |
| MODIS above 500 m a.s.l. | - | 2.4 | | - | 0.11 |
| MODIS all elevations | - | 2.6 | | - | **0.04** |


Figure 4a shows the relative change in annual SCF over Iceland as estimated from MODIS observations over the period 2001-2021. Figure 4b shows areas where the trendline is statistically significant ($\alpha = 0.05$) for both MODIS and in situ observations (SNC = 4), in situ observations shown as symbols. Blue regions and markers show areas where the SCF had increased significantly, and the red areas with decreasing SCF. The results show that many areas in Iceland have experienced a significant

change in the local SCF, both as estimated from MODIS data and from manned snow cover status observations over the period 2001-2021. Most of these areas have experienced an increase in SCF, especially the eastern highlands and the mountainous regions of Northern and Northwestern Iceland. A few areas showed significant decreases in the SCF and most of those were located at the termini of the country's major outlet glaciers, whose retreat has been well documented (Aðalgeirsdóttir et al., 2020; Hannesdóttir et al., 2019; Hauser & Schmitt, 2021) or in coastal areas. The areas where the largest change in SCF is

observed from MODIS data over the period 2001-2021 are those where geophysical surface changes have occurred, e.g. due to glacial retreat. All manned observations sites where a decrease in SCF or *SND* had occurred over the period were all located at low elevation in coastal areas except for one.

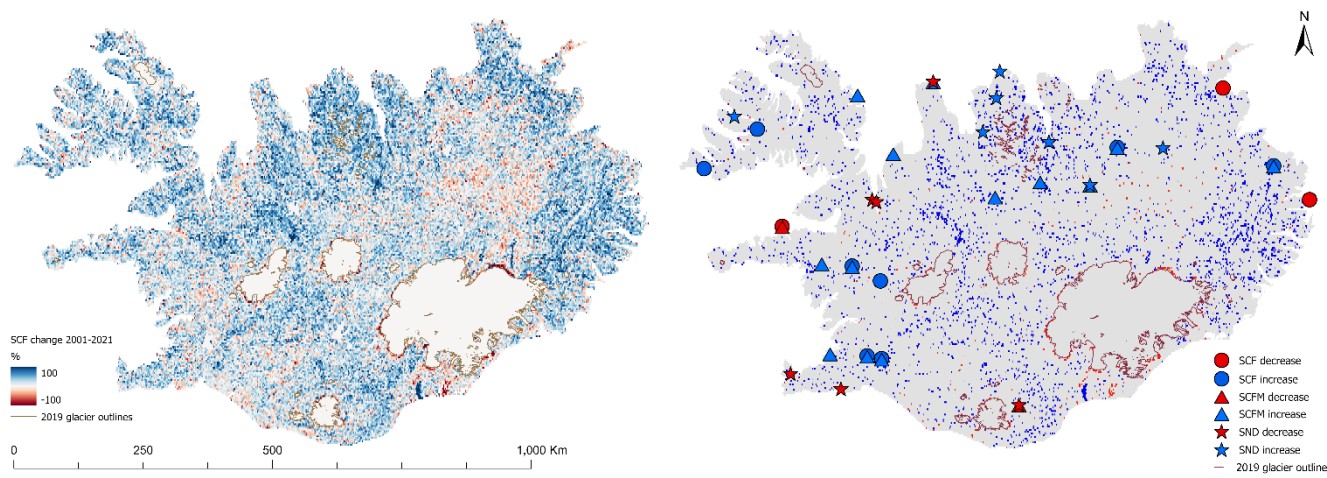

**Figure 4 Left panel (Fig 4a): Percentage relative change in annual SCF over Iceland as estimated from MODIS over the period 2001-2021. Right panel (Fig 4b): areas where the SCF trendline is statistically significant ($\alpha = 0.05$) for both MODIS and in situ observations (SNC = 4), in situ observations are shown with symbols. 2019 outlines of glaciers and the ice divides of their major outlet glaciers are shown with black lines(Hannesdóttir et al., 2020).**

### 3.2 Projected Seasonal Snow Conditions

Figure 5 shows the results of the simulation of daily snow conditions in Iceland for the period 1950-2100 for both Representative Concentration Pathways (RCP) RCP4.5 and RCP8.5. 5a shows the average winter SWE across Iceland and 5b shows the simulated average annual SCF along with in situ and MODIS derived SCF estimates.

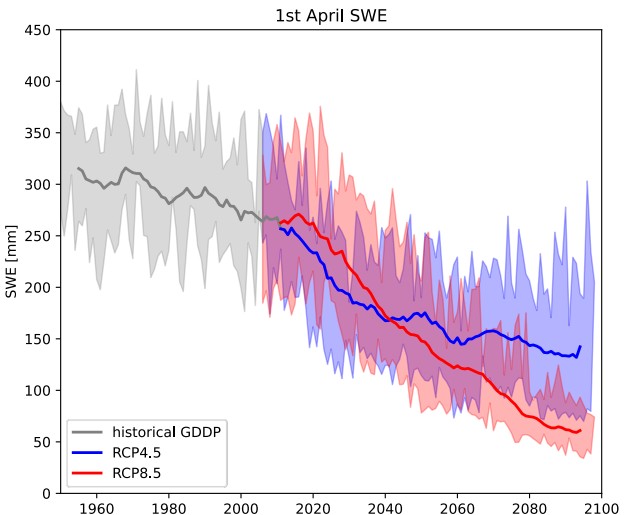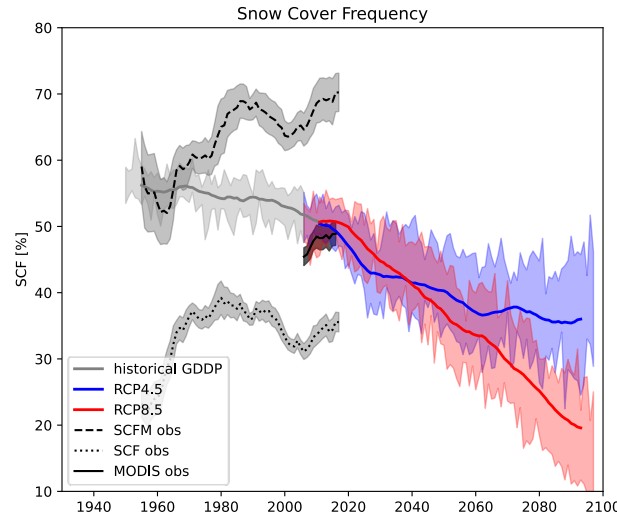

**Figure 5 Left panel (Fig. 5a): simulated average winter SWE across Iceland for both RCP4.5 (green) and RCP8.5 (red). Right panel (Fig. 5b): simulated average annual SCF across Iceland as projected by RCP4.5 and RCP8.5 compared to observations from monitoring stations of mountain and local snow cover status as well as the MODIS snow cover products. The shaded area represents the upper and lower quantiles of the ensemble simulations and one standard deviation from the mean of the observations. The solid line shows a 10-year moving average.**

Figure 5 shows that both SWE and SCF are expected to decrease in Iceland over the course of the 21st century and that this decrease has been ongoing throughout the study period. Figure 5b shows that MODIS derived SCF estimates over the period of 2001-2021 fit well with the simulated values. In situ observations of local and mountain snow cover status (SNC or SNCM > 2) fall below and above the simulated averages, respectively, as expected. However, although observed and simulated SCF estimates fit well with the magnitude and variability of each other, their trend is opposite. The observational data show an increasing SCF trend while the simulations show a decreasing trend over the historical period 1950- 2020.This pattern of opposing trends is also observed in terms of snow magnitude as the simulated SWE estimates show a decrease in SWE, whereas the observed snow depth measurements (shown in Figure 2) show a significant increase ($p = 1.54 \times 10^{-5}$) over the period 1930-2021. The results also illustrate the substantial natural climate variability in Icelandic snow conditions.

The results in Figure 2 show a positive trend for temperature and precipitation in Iceland over the period 1950-2021. Increasing temperatures result in enhanced snowmelt, which is apparent in a flat or decreasing SCF in coastal regions (shown in Figure 4), whereas at higher elevation the increased precipitation enhances winter snow accumulation leading to higher SCF despite the enhanced snowmelt during summer, leading to a countrywide increase in average SCF. This effect of increased snow cover at high elevations can be expected to persist until temperatures have risen above freezing for a considerable portion of the winter at the highest elevations as well, after which snow-cover is expected to decrease at all elevations. Due to variability in the Icelandic landscape and topography this effect should be more apparent when simulated at a finer spatial resolution.

Recent studies have suggested a regional cooling in the ocean temperatures surrounding Iceland due to changes in the thermohaline circulation in the North Atlantic Ocean (Caesar et al., 2018). This regional cooling, which has been connected to a temporary slowing of glacial ablation in Iceland (Noël et al., 2022), would explain the opposing trends observed in Figure 5 as this regional cooling is poorly represented in the CMIP5 models used for the snow simulations in this study. The North-Atlantic cooling trend is projected to halt around 2050 given the results of the Community Earth System Model version 2 (Danabasoglu et al., 2020).

## 4. Discussion and Conclusion

The analysis of snow observations showed a significant increase in both Snow Cover Frequency (SCF) and 1st April SWE, both as estimated from in situ observations over the period 1930-2021 and from observations from the MODIS instruments on NASA's Terra and Aqua satellites over the period 2001-2021. The MODIS observations were comparable with in-situ observations of both local and mountain snow cover status. The results also revealed a large natural variability in snow conditions, which was expected due to the sensitivity of the Icelandic climate to fluctuations in large scale atmospheric and ocean circulations in the north Atlantic region (e.g. Hanna et al., 2004; Massé et al., 2008).  The results showed a significant increase in average annual snow depth over all stations for the period 1930-2021.

Simulated Snow Cover Frequency, SCF, was consistent with SCF estimates from both MODIS and in situ observations for the historical period, although the simulated trend was opposite to the trends in both observational datasets. The simulations show that SCF is expected to decrease significantly over the projected period, 2006-2100 especially below 500 m a.s.l. where snow cover is expected to become a rare occurrence by the end of the period, given the RCP8.5 emission scenario. The simulated SWE shows a significant decrease in SWE over the period 1950-2021 whereas average annual SND from all IMO stations has a positive trend over the same period. The results show that the water storage in Icelandic winter snowpack could decrease by about half or 3/4 under the RCP4.5 and RCP8.5 emission scenarios, respectively, over the period 1950-2100.

The results of this study suggest that the increased SCF in Iceland, observed both from remotely sensed and in situ data, is associated with increased precipitation causing a more frequent and thicker snowpack which persist longer, despite enhanced melt rates. This is consistent with Bjornsson et al. (2018) who found annual precipitation to have increased by about 10% during the period 1980-2015. This increasing trend was also observed by Gunnarsson et al. (2019) which used multi-source satellite remote sensing data to show that there had been an increase in snow cover in Iceland for all months except October and November over the period (2000-2017).  The simulated snow conditions are also in agreement with previous projections of a decrease in snow cover frequency and snow mass across Iceland, as rising average temperature causes spring melts to begin earlier and autumn snow cover to occur later (e.g. Johannesson et al., 2007).

The results presented in this study deserve further investigation. Observations of snow conditions reveal a large natural variability which may be affected by large scale circulations in atmospheric and ocean circulations in the northern Atlantic as well as global temperature changes. The observations of both snow cover and snow depth indicate an increasing trend in these

variables over the historical period whereas simulated snow conditions predict a decrease in both over the course of the present

century, the extent of which is dependent on future emission scenarios. The observed increases in SCF and SWE could be part of natural climate variability induced by low frequency cyclical climate patterns, or by a small amount of extreme weather events. The causes and the impacts of these changes to Icelandic ecology and society should be better understood as future changes to snow conditions will impact the hydrological cycle, which will further affect the local ecology, hazard assessments, water resources management, and hydropower production in the country.

**5. Author Contribution**

DE and SMG designed the experiments. DE developed the code and performed the analysis and prepared the manuscript. DE and AG gathered, assessed, and prepared the data. SMG, AG and OGBS reviewed the manuscripts and provided significant consultation and contributions throughout the work.

**6. Competing Interests**

The authors declare that they have no competing interests.

**7. Code/Data Availability**

All data used for the analysis in this study are freely available and were accessed either through the Google Earth Engine database or by direct correspondence with the data provider. The datasets used in this study and their source literature and links are provided in Table 3. The code for the snow model and/or the remote sensing analysis can be made available upon

request.

Table 3 Underlying research datasets used in this study

| Dataset | Purpose | Reference | DOI: |
|---------|---------|-----------|------|
| NASA NEX-GDDP | Model forcing | (Thrasher et al., 2012) | doi:10.5194/hess-16-3309-2012 |
| GLDAS-2 | Parameter estimation | (Rodell et al., 2004) | DOI:10.5067/E7TYRXPJKWOQ |
| Arctic Permafrost Extent (WGS43261) | Parameter estimation | (Brown et al., 2002) | https://doi.org/10.7265/skbg-kf16 |
| IslandDEM | Parameter estimation and analysis | - | https://www.lmi.is/is/landupplysingar/gagnagrunnar/nidurhal |

| MYD10A1.006 MOD10A1.006 | Remotely sensed snow observations | (Hall et al., 2016) | https://doi.org/10.5067/MODIS/MOD10A1.006 https://doi.org/10.5067/MODIS/MYD10A1.006 |
|---|---|---|---|
| IMO snow observations | In situ snow observations | - | fyrirspurnir@vedur.is |

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
