# Peer review of "Observed and Predicted Trends in Icelandic Snow Conditions for the period 1930-2100"

_EGUsphere, 2022_

## Referee Comment (RC2)

**Review of Eythorsson et al. (2022; doi:10.5194/egusphere-2022-590)**

**General comments**

This manuscript presents an impressive multi-faceted analysis of the historical and (possible) future snow cover on Iceland based on in-situ observations, remotely sensed data, and climate simulations. The fact that much of the analysis, including the snow modeling, seems to have been carried out in the cloud was particularly novel to me and should be of great interest to the wider cryospheric community. Among the main results of this study, historical observations show an increasing trend in snow cover frequency (SCF) over Iceland during the last decades whereas projections from the simulations indicate that SCF will decrease substantially in the future especially under the more aggressive emissions scenario. This could suggest a threshold effect related to changes in precipitation phase under a warmer and wetter future climate.

While the manuscript was generally well written it is quite brief and parts of it felt a bit rushed. For example, the text could have benefited from more examples of recent related work on remote sensing and modeling for snow cover reanalysis (e.g. Alonso-González et al., 2021; Liu et al., 2021) and projections (Fiddes et al., 2022). The authors could also have taken advantage of their efficient conceptual snow model combined with downscaling (see Fiddes et al., 2022, and references therein) to conduct simulations at resolutions nearer to the hillslope scale (Fan et al., 2019) to try to better capture the spatial variability of the snowpack. Moreover, it did not seem as though the parameters in the SNOW-17 model were well calibrated using local information. This is a lost opportunity, as the MODIS data would have been a good candidate to calibrate many of the parameters (e.g. Fiddes et al., 2019; Alonso-González et al., 2021; Liu et al., 2021) such as the gauge under-catch factor and the range on the melt factor. This could have helped to constrain uncertainty in the SNOW-17 model and the forcing. It was also odd that the GDDP forcing data was treated almost as a reanalysis dataset during the historical period rather than as (downscaled and bias-corrected) climate model output. In particular, this data is meant to represent the the correct climate on decadal timescales but not the day to day weather or even inter-annual variability. As such it is not fair to compare observations directly to SNOW-17 forced with this dataset. For such an exercise it would have made more sense to use actual reanalysis data, such as ERA5, to force the SNOW-17 model. Such simulations could be compared directly to the observations. Furthermore the climate (i.e. decadal moving averages) from the ERA5-driven simulations could be compared to the GDDP-driven simulations in the historical period (1950-2021) to gauge the performance of or bias correct the latter (Fiddes et al., 2022). I also had some concerns about the way the MODIS data was processed, particularly that no gap-filling was conducted before calculating SCF, despite previous work on such methods by some of the authors (Gunnarsson et al., 2019). This should at least be justified in the text.

As such, I suggest that this manuscript should undergo major revisions. Nonetheless, I would like to commend the authors for this valuable work that fits well within the scope of *The Cryosphere* and I encourage them to address these general comments and the specific comments below.

**Specific comments**

- L9: Change *"remote sensing observations"* to *"remotely sensed observations"* (replace all).

- L10: Use either *"General Circulation Model"* or *"Global Climate Model"* (preferably the former) but not *"Global Circulation Model"*.

- L11: One immediately wonders why the CMIP6 version of this dataset was not used. This is not a criticism per se, but the existence of a newer version of the NEX-GDDP dataset should at least be mentioned somewhere in the manuscript.

- L13: Representative Concentration Pathway 4.5 (RCP4.5) indicates a scenario where the radiative forcing will be 4.5 Wm$^{-2}$ at the end of the century. So please use the *"."*, it's RCP4.5 not RCP45. The same holds for RCP8.5.

- L13: Change *"Snow17"* to *"SNOW-17"* in line with Anderson's naming convention (replace all).

- L20: Suggest changing *"climate"* to *"micro-climate"* also change *"significantly"* to *"strongly"* since significant has a specific statistical meaning in your manuscript.

- L21: Change *"correlate to changes in"* to *"are highly correlated with changes in"*.

- L30: Change *"at least mid"* to *"at least the middle of the"*.

- L33: Suggest changing *"duration of snow cover"* to *"snow cover duration"* which is perhaps more widely used (e.g. Notarnicola, 2020).

- L34: Best in what sense? A more qualified statement would be to say that a good balance of spatial and temporal resolution as well as temporal coverage is a valid reason for using MODIS. You could also consider citing a paper (e.g. Aalstad et al., 2020, and references therein) that evaluates satellite-based snow-covered area products from MODIS and give an idea of what uncertainty you would expect at the pixel scale.

- L40: Change *"and is"* to *"which is"*.

- L41: Change *"earths"* to *"Earth's"*.

- Fix typo *"thatasc"*.

- L45: This could be a good place to mention other studies that have performed detailed future snow cover projections under different scenarios (e.g. Fiddes et al., 2022, and references therein).

- L47: Change *"Snow Covered Area"* to *"Snow-Covered Area"*.

- L49: It would be natural to mention that SNOW-17 is a conceptual model built around the degree day (also known as the temperature index) approach. Currently this is not mentioned anywhere in the manuscript. Since it is a degree day model, it is not only more efficient but it also has less requirements in terms of forcing data than full energy balance models. On the other hand, given that the entire energy balance is lumped into a single term, one would think that calibrating the degree day factor is critical for the model.

- L51: Change *"and it"* to *". It"*.

- L55: Change *"calculated"* to *"estimated"* since this is an approximation.

- Section 2.1.2: The formulation in this section should be improved. A reader that is unfamiliar with these MODIS snow cover products could think that 'NDSI_Snow_Cover' is meant to measure Fractional Snow-Covered Area (FSCA) directly. This is not the case, instead the valid pixels in the 'NDSI_Snow_Cover' field merely contains the NDSI value (scaled by 100, see e.g. Riggs and Hall (2020)) for pixels that could possibly contain snow (positive NDSI) or a value of 0 for pixels that probably do not contain snow (NDSI$\leq 0$) and have passed various screens (not deemed to be cloudy etc...). This is readily verified by comparing it to the 'NDSI' field which will have equal values once negative values are set to zero and missing (cloudy etc...) pixels are masked out. The NDSI is of course related to the FSCA of a pixel, but they are not identical and converting NDSI to FSCA usually involves some form of transformation. In particular, it is often the case that even pixels with an intermediate positive NDSI value (say equal to 0.6) can be fully snow-covered (FSCA=1). It would be worth making this clear to the reader and discussing the commonly used linear relationship between the two (Salomonson and Appel, 2006; Fiddes et al., 2019; Alonso-González et al., 2021), along with some uncertainty estimates form the literature (Aalstad et al., 2020, and references therein). Indeed an advantage with the C6 (versus C5) MODIS snow cover products is that users can customize the NDSI-FSCA relationship for their own use case (see the MODIS Snow Products Collection 6 User Guide user guide). Work with Sentinel-2 (Gascoin et al., 2020) has shown that other types of NDSI-FSCA relationships (e.g. sigmoid) can also perform well

- L73: How much did glacier outlines change during the MODIS era (2001-2021)? One would assume that this effect is small, but make it explicit that you assume these outlines to be constant in time for this period.

- L78: It would be nice to have at least a rough idea of what areal scales local snow cover and surrounding mountain snow cover represent? Moreover, the term *"snow cover"* is very general (see the NSIDC glossary) and a bit vague in this context in that it can implicitly refer to many different snow variables. You clarify your

use of the term a bit later (L82) as being some kind of a snow cover classification (snow/patchy/no-snow) that lies between binary snow cover (snow/no-snow) and FSCA. I would still recommend calling it something more specific than *"snow cover"* such as *"snow cover status"* or similar.

- L91: This sentence seems unnecessarily convoluted. Based on your classification system, an equivalent but more concise way to define *"snow-covered ground"* would be with the inequality SNC$> 0$ (or equivalently SNC$\geq 2$ since there is no class SNC$= 1$) and SNCM$> 0$ for local and surrounding mountain snow cover, respectively.

- L99: This is implicitly assuming that NDSI$> 0$ corresponds to FSCA$> 0$. While this is often true, in the sense that low but positive NDSI in a pixel can be due to a patchy snow cover, you may have some false positive 'snow-covered' pixels as a result of this, since some non-snow surfaces may also have a low but positive NDSI. Did you look at the sensitivity of your results to this threshold, for example by trying a threshold of NDSI$> 0.1$ instead which may be less sensitive to such false positives. Moreover, you could quantify the false positive rate by for example comparing your MODIS-based snow cover status classification to that from the station measurements. This could help to calibrate the NDSI threshold. Perhaps it turns out that 0 is good choice of threshold threshold, but testing this can help to strengthen your analysis and the resulting conclusions.

- L102: By excluding December and January from the analysis your SCF may end up incurring a negative bias (i.e. be underestimated). In particular, the months of December and January often have a higher SCF than many other months in the year. Since this is likely to be the case, your results may (at least on average spatially) end up being closer to the true annual SCF if you gap-filled your observations using either simple interpolation techniques or more sophisticated algorithms. More generally, applying gap-filling could also make your results more robust to a potentially uneven temporal distribution of cloudiness throughout the year. Given the large overlap in authorship, I found it surprising that you did not use the gap-filling method of Gunnarsson et al. (2019) which has already been applied successfully over your domain.

- To properly test the performance of SNOW-17 given weather (rather than climate) forcing I would strongly encourage the authors to also run SNOW-17 for the historical period (1950-2021) with forcing from the ERA5 reanalysis. Note that this product is also available on Google Earth Engine, so it should be relatively straight-forwards to extend your analysis. This would be a fairer model simulation to compare with in-situ observations, which (disregarding scale-mismatches) experience (roughly) the same weather as in ERA5 rather than the weather simulated by GDDP which aims only to produce the correct climate (i.e. weather statistics). This ERA5 simulation would also a reference at roughly the same spatial scale as your existing GDDP-forced simulations that you could use to validate the climatic evolution (i.e. at the decadal timescale) of the snowpack in the historical period.

- L117: Change *"of trend"* to *"of trends"*.

- L119: It could be more instructive to cite studies that have applied similar trend analysis methods for snow cover such as Yılmaz et al. (2019) and Notarnicola (2020).

- L120: This statement is misleading. When doing null hypothesis significance testing you are *not* testing the alternative hypothesis. You are not even really testing the null hypothesis. For more about the caveats of significance testing and the error of the transposed conditional see Ambaum (2010). To be more specific, I would urge the authors to explain what their $p$-values actually quantify.

- Figure 2: Please improve the resolution and visibility of this figure. For example, text should not be visibly pixelated and it should be easy to differentiate markers without zooming excessively.

- Suggest changing *"ensemble average"* to *"ensemble mean"* which is perhaps more commonly used.

- L126-130: This is essentialy a repetition of the figure caption. Please shorten considerably, this should only briefly describe the results that the figure shows.

- L141 Change *"trended upward"* to *"a positive trend"*.

- L142 According to your earlier description the inequality describing full *and* patchy snow cover should be SNC$\geq$ 2 not SNC$>$ 2.

- L146 Instead of speculating about what the increase in precipitation could lead to in terms of snow accumulation, you could analyze the changes in snowfall and rainfall rates to a first order by applying simple air temperature-based thresholds to delineate precipitation phase.

- Table 2: Please use $\times$ for multiplication not $*$. Also fix *"of p-values Statiscially significant..."* in the caption. Moreover, I would recommend setting all the text in the table to normal font apart from significant *p*-values which can be in bold.

- Figure 3: Please center the colormap around 0.

- L176: Once more you are just repeating the caption. Please shorten considerably and avoid redundant text as much as possible.

- Figure 4: To make the long-term weather observations and the GDDP-driven climate simulations of SCF more comparable it would be natural to instead plot the decadal moving mean (serving as a low pass filter) of the observations. The corresponding moving standard deviation would then help to visualize the observed internal climate variability. In that way, you would be more fairly comparing observed climate to the simulated climate. In particular, as previously alluded to, GDDP is not a reanalysis so we can not expect it to reproduce the correct weather.

- L190: Remove *"fig"*.

- L192: Why aren't the observed and simulated trends compared for an overlapping period, such as 1950-2021?

- L204: What does *"increase in snow cover"* mean here? Are you referring to frequency, duration, depth, SWE? Please be more specific.

- L205: This is not shown clearly in Figure 2, although maybe it will be easier to see when the Figure is sharpened. I would also recommend to add a panel to Figure 2 containing a scatter plot that compares the MODIS SCF to in-situ SCF observations for both local and mountain snow cover.

- L210: Please make it clear here that although the simulated SCF magnitude is comparable to the observations, the trend is in the opposite direction.

- L213: It's of course not straightforwards to directly compare SWE to snow depth given variations in snow density. Moreover, Figure 2 shows annual mean observed snow depth whereas Figure 4 shows simulated April 1$^{st}$ SWE. Please clarify what you are comparing here. While the unrepresentative (mainly) low-lying station locations can account for some of the differences, you could perhaps make the simulations more comparable to the observations by excluding higher elevation grid cells.

- L218: Once more *"increase in snow cover"* is vague, please explain what is meant by this.

- L224: Change *"predictions which forecast"* to *"projections of"*.

- L229: Change *"parameters"* to *"variables"*.

- L230: Although the acronym GHG is well-known, this is the first time it is used in this manuscript. Consider writing it out in full or just change *"GHG emissions"* to *"emission scenarios"*.

- Code/Data availability: Please provide references (with DOIs) or link to all the datasets used.

I'm looking forwards to reading the revised manuscript,
Kind regards,
Kristoffer Aalstad

**References**

Aalstad, K., Westermann, S., and Bertino, L.: Evaluating satellite retrieved fractional snow-covered area at a high-Arctic site using terrestrial photography, Remote Sens. Environ., doi:10.1016/j.rse.2019.111618, 2020.

Alonso-González, E. et al.: Snowpack dynamics in the Lebanese mountains from quasi-dynamically downscaled ERA5 reanalysis updated by assimilating remotely sensed fractional snow-covered area, Hydrol. Earth Syst. Sci., doi:10.5194/hess-25-4455-2021, 2021.

Ambaum, M. H. P.: Significance Tests in Climate Science, Journal of Climate, doi:10.1175/2010JCLI3746.1, 2010.

Fan, Y. et al.: Hillslope hydrology in global change research and Earth system modeling, WRR, doi:10.1029/2018WR023903, 2019.

Fiddes, J., Aalstad, K., and Westermann, S.: Hyper-resolution ensemble-based snow reanalysis in mountain regions using clustering, Hydrol. Earth Syst. Sci., doi:10.5194/hess-23-4717-2019, 2019.

Fiddes, J., Aalstad, K., and Lehning, M.: TopoCLIM: rapid topography-based downscaling of regional climate model output in complex terrain v1.1, Geosci. Model Dev., doi:10.5194/gmd-15-1753-2022, 2022.

Gascoin, S. et al.: Estimating Fractional Snow Cover in Open Terrain from Sentinel-2 Using the Normalized Difference Snow Index, Remote Sens., doi:10.3390/rs12182904, 2020.

Gunnarsson, A., Garðarsson, S. M., and Sveinsson, O. G. B.: Icelandic snow cover characteristics derived from a gap-filled MODIS daily snow cover product, Hydrol. Earth Syst. Sci., doi:10.5194/hess-23-3021-2019, 2019.

Liu, Y., Fang, Y., and A, M. S.: Spatiotemporal distribution of seasonal snow water equivalent in High Mountain Asia from an 18-year Landsat–MODIS era snow reanalysis dataset, The Cryosphere, doi:10.5194/tc-15-5261-2021, 2021.

Notarnicola, C.: Hotspots of snow cover changes in global mountain regions over 2000–2018, Remote Sens. Environ., doi:10.1016/j.rse.2020.111781, 2020.

Riggs, G. and Hall, D.: Continuity of MODIS and VIIRS Snow Cover Extent Data Products for Development of an Earth Science Data Record, Remote Sens., doi:10.3390/rs12223781, 2020.

Salomonson, V. V. and Appel, I.: Development of the Aqua MODIS NDSI Fractional Snow Cover Algorithm and Validation Results, IEEE Trans. Geosci. Remote Sens., doi:10.1109/TGRS.2006.876029, 2006.

Yılmaz, Y. A., Aalstad, K., and Sen, O. L.: Multiple Remotely Sensed Lines of Evidence for a Depleting Seasonal Snowpack in the Near East, Remote Sens., doi:10.3390/rs11050483, 2019.

---

## Author Comment (AC1)

| Referee Comment | Author Response |
|---|---|
| **General comments:** | |
| This paper provides an interesting analysis of past and future snow conditions in Iceland. The topic is very relevant for the journal The Cryosphere and the authors provide some interesting tools and data to support their research. The study provides a novel contribution to Icelandic snow conditions by combining and comparing in-situ observations, remote sensing estimates and climate model simulations, which had not been done before. The geographical situation of Iceland in the North Atlantic with its maritime and cold climate makes the findings interesting for the scientific community. Nevertheless, in my opinion there are quite a few major concerns that the authors should address before this paper can be published in TC. I believe addressing these concerns would highly improve the quality and especially the trust in the findings. I see the potential for a very good quality and highly-relevant paper after these issues are addressed.

I give a detailed description of the major concerns in the section below. In summary, I first of all believe the authors should clearly state the novelty and relevance to the scientific community of their findings. While this is clear to me as I mention above, they need to make it clear in the introduction. While I liked that the paper is concise and to the point, I found the description of the modelling part too short, with key details and descriptions missing. The statements about their calculated trends are a point of major concern. I think the authors should be more open about what they observe in the figures. While increasing trends in snow cover frequency are observed, decreasing trends in snow cover frequency are simulated. These simulations are then the basis of the conclusions that snow cover frequency will decrease in the future. It is ok if observations and simulations disagree, I find it would be interesting to know the reasons why, rather than claiming | The authors express their gratitude to the referee for thorough and insightful review. We have addressed all general and technical comments and which we believe has resulted in a significantly improved article.

We agree that the novelty of the study was not clearly stated in the previous version. A description of the novelty of the study has been added to the last paragraph of the introduction section as suggested by the referee.

We also agree that the section describing the modelling could be expanded and we have revised and extended the entire modelling section as per the referee's suggestions.

The section on the trends in observed and simulated snow conditions has been revised and clarified and a discussion on the physical basis for the observed trends has been added.

The technical corrections provided by the referee have all been addressed including the revision of figure and table caption and descriptive text.

The description of data availability in Section 7 has been revised and expanded

Please see specific responses below. |

something that might be wrong. This could generate mistrust in the findings.  The authors should address the technical corrections that I state below, as well as some parts of the text that are literally repeated in the text and in the captions. Finally, it is essential that the authors provide a statement on how their underlying research data can be accessed (as per TC data policy), since at the moment it is only stated that "the data is available".

I apologise for the long review and I positively encourage the authors to address my comments. I look forward to seeing a revised version of the manuscript, which I am sure it will be better and suitable for publication.

**Specific comments:**

| | |
|---|---|
| Introduction: The introduction is generally good and concise, but in my opinion it is missing information on the importance of studying changing snow conditions in Iceland and the novelty that this study brings. The first paragraph starts with a general introduction to Icelandic climate and previous studies in Iceland. I miss a closing sentence stating why focusing on changing snow conditions in Iceland is important (e.g. snow is a major component of the water balance in Iceland, the geographical position of Iceland and the North Atlantic influence…). The second paragraph introduces remote sensing, snow cover variables, and the snow modelling. Then the objective is "to analyse observed trends and predict development of snow conditions in Iceland". I can't clearly see from the text where is the knowledge gap that the authors are filling with this study. Have future snow conditions not been analysed for Iceland before? Or has this modelling approach not been used before? Is it the combination of observed and predicted trends? What is the novelty? I do see the novelty and importance of the study, I just think the authors should state that clearly. | Good points.

Added to first paragraph to highlight the importance of the study:

"*Understanding of future expected changes to snow in Iceland is important for water resources management as it constitutes a significant portion of the regional hydrological cycle, especially in the interior highlands where the majority of the country's energy production occurs, in hydropower plants developed on glacial rivers.*"

Added to the third paragraph to highlight the novelty of the study.

*"The novelty of this study is the analysis of an extended dataset of in-situ records of snow conditions in Iceland combined with reliable remotely sensed dataset of snow conditions in the area and the comparison of these observations with snow conditions simulated using a trusted snow model run with downscaled and bias corrected temperature and precipitation estimates from an ensemble of 21 the CMIP climate models ensemble on a freely available, cloud based, parallel computing platform.*" |

| | |
|---|---|
| To be consistent between sections 2.1 and 2.2, I would change the order of subsections in 2.1, so that In-situ snow observations go first (2.1.1) then remote sensing (2.1.2) and then Climate Data (2.1.3). | Good point. The order of these sections has been changed. |
| Line 100: What if for a specific year there are only valid observations in summer, or only at a different time of the year than other years? How would you handle that, is there a minimum threshold of valid observations, or a defined distribution over the year that the valid observations must follow? | It is inherent with satellite observations that they are limited by factors such as cloud cover causing an unequal distribution of observations within years. In this study we used observations from the MODIS instruments that record observations over Iceland twice daily. We did not perform gap-filling of the dataset to prevent introducing another source of uncertainty and thus based the analysis on the observations themselves. As the aim of the study was to investigate long term trends in snow cover interannual variability within specific pixels not considered. |
| Snow modelling: The description of the modelling part is too short. Although I understand some things are explained in the cited references, there should be a minimum model description with key processes. How are the parameters estimated? Table 1 says they come from Eythorsson et al. 2021, but this reference is not in the refence list! What are the good things of this Snow17 model? What are its limitations? What resolution is used? (Only found out that Snow17 model output has the same resolution as the forcing GCM in line 176 in the results). Why can't the model be run at higher resolution, given that other model parameters are probably available at higher resolution? What is the simulation period? I only found out in the discussion that it was 2006-2100. This should be explained here. Why starting in 2006? | We agree that the description of the snow modeling was short and confusing, this section has been restructured and revised. The background literature on the Snow17 model has been expanded in the introduction.

Please note that the simulation period was 1950-2100, as is now clearly stated in the first sentence in Section 2.2.3. |
| Line 114: Why use 1st of April SWE? See Nolin et al., 2021. | We use the 1st April SWE as it has been historically favored as an effective index for streamflow forecasting (Bohr and Aguado, 2001) and is thus a comparable metric with prior studies in the field. We agree with the referee and Nolin et al. 2021 that it doesn't represent mid-winter melt events and is thus problematic for future streamflow forecasting as the probability of these events increase, which is why we also included the SCF metric, which is recommended by |

| | Nolin et al. 2021 as a new snow metric for a warming world. This has been clarified in the text in section 2.2.3 which has been rewritten. |
|---|---|
| Lines 117-118: A bit confusing. What are the "calculated time series" and the "distributed observations"? Sen's slope calculated a slope, not a significance. The significance is estimated with the MK test. Please rephrase and make it clear. | Agree. This has been rephrased as: "The statistical significance of the trend in the time series of in situ observed mean annual SCF, SND was estimated using the Mann-Kendall trend test and the significance of trends in MODIS observed SCF was estimated using Sens's estimator of slope method." |
| Figure 2c: As far as I understand it, each point here is the average snow depth from all available in-situ observations over Iceland for a specific year. However, the authors selected all stations with at least 20 years of available observations in the period 1930-2021. In the case where e.g. in the 1940-1960 there were more station observations available at lower elevations compared to 2000-2020, it could be that the Iceland average was lower in certain years merely due to different distribution of the availability of measurement stations. I am confident this is not the case, but this has to be shown, otherwise the results could be completely wrong. Perhaps a plot showing the availability of ALL stations observations against elevation or against mean snow depth. This could be shown together with Figure 1. | Good point. A plot of the number of observation stations against the mean snow depth observed has been added to Figure 1. |
| Line 147: I don't think melt rates is the correct word here, since melt rate is the rate at which snow melts, but not the total amount of melt or the duration of the melt season. I suggest "offset the increased winter snowmelt and shortening of the snow cover duration associated with temperature rise". Also in line 220 change "melt rates" for "snowmelt". | Agree, thanks for this suggestion. This has been changed in the manuscript. |
| For all figures and tables in the results section, the text in the figure captions is also written in the section text: e.g. Lines 148-150 are the exact same lines as 154-155 which is the Table 2 caption; same in lines 160-161 which are the exact same lines as 164-165. Lines 178-179 and 184-185 are also the same. This is not good practice. | Good point. The captions and texts describing figures has been revised for all tables and figures. |

| | |
|---|---|
| Caption should give a title to the figure/table and explain the details of the figure that are not self-explanatory. The text should explain what the results in the figure show (a decreasing trend, a high value for X, etc). Text should not be repeated in the caption and the text, let alone a copy-paste… Please change this for all figures and tables. | |
| Figure 3a: The colour bar should have the white colour at 0, so that no trend is shown as white in the map. At the moment it looks like the 0 is at +3, which could give a wrong impression in the figure. Also somewhere on the paper please explain why there no observations on the glacier (is it a limitation from MODIS?). | Good point. The figure has been revised to make clear that the white color is at 0. The observations do cover the glaciers and the SCF change on the glaciers is 0 as would be expected. This is now more clear with the revised figure. |
| Figure 3b: It is not clear to me what the difference is between symbols and non-symbols. As I understand it, significant MODIS trends are shown all over Iceland, and significant IMO station trends are shown additionally as a symbol? Please provide a clearer explanation if so. | The referee is correct. We have revised the text to clarify that symbols represent the IMO stations |
| Figure 4: For the historical period, why is it shown in red colour as if it was the RCP8.5? If there is a reason, explain it. If there isn't one, then the historical period up to 2006 should be displayed with a different colour. | Good point. The figure has been revised to show the historical period in grey. |
| Trends: There is something inconsistent in the trends and that is in my opinion wrongly explained in the text. Observations of IMO stations and MODIS show increasing SCF trends for the historical period. This seems consistent within all the results and literature shown based on historical observations (except see my comment about Figure 2c above). However, Figure 4 clearly shows a decreasing trend in SCF (and SWE) for the historical period, based on simulations. However, the authors state that "the simulated estimates of average SCF shown in Fig. 4b are in line with MODIS observations over the period 2001-2021" (see line 191). This is not what I see in Fig. 4b: even though the order of magnitude of the SCF values is good (good fit between observed and simulated), the trend is opposite. Table 2 claims an increasing trend in SCD for MODIS, while the simulations | Good point, we agree that the text was confusing as to discerning between the consistency of the trends and the „fit" between observed and simulated values. We agree that the trend is indeed opposing between the simulations and the observations.

The text describing Figure 4 has been rewritten and expanded to clarify this point including the addition of a paragraph on the likely reason for the opposing trends.

The authors agree that the opposing trends revealed in this study are important and deserve further investigation in future research and publications. |

| | |
|---|---|
| in Figure 4 show a decreasing trend for that same period. This tells me there is something wrong either with the observations or the simulations, or simply with the text. The increasing vs decreasing trend problem is even more apparent for the historical period (1930-2020). IMO observations show increasing SCF over 1930-2020, while the model simulations show a decreasing trend for 1930-2020. So why is the simulation showing a decreasing trend in the historical period? And how could this impact the statements that are made about the future regarding SCF? It does not generate much trust in the future projections. I think it could be very interesting to look into the reasons behind this disagreement, without this being necessarily a bad thing for the paper or for the results. There might be an explanation and the authors should investigate it further, providing more convincing results and discussion. | |
| Data availability: The authors state that "all data are freely available", but there is no information whatsoever on where is the data available. Please provide all details about the data and where to find them (IMO data, MODIS, NEX GDDP, etc), with links. | Section 7 has been revised and extended. A table has been added with links to all underlying data used. |
| **Technical corrections:** | |
| Check the references (e.g. in Line 22 it should be "Eythorsson et al., 2018"). Please revise all references in the text are in the reference list and vice-versa. | The references have been checked and revised. |
| Line 24-25: Better to use change per decade than per century, since the period is 1980-2016. | Agree, this has been changed. |
| Line 38: Remove "from" | Removed |
| Everywhere: Add dots in RCP scenarios, it is RCP4.5 not RCP45. Same with RCP8.5. Also, why did you choose these scenarios? | Correct. These dots have been added to the rcp scenarios.

 Reasoning for the RCP scenarios chosen has been added to section 2.2.3 as follows.

 *„These scenarios were chosen to represent both a „business-as-usual" scenario (RCP8.5) and a stabilization scenario (RCP4.5) where anthropogenic climate forcing are assumed to be stabilized by the end of the century."* |

| | |
|---|---|
| Line 40: Remove e.g. from (e.g. Nolin et al., 2021). E.g. is only needed to state "for example", but in this case it is Nolin et al. who defined SCF. Same in line 41. It is correct in line 44. | Changed |
| Line 43: wrong typed word "thatasc". I guess it should just be "that". | Correct, this has been changed. |
| Line 116: I think it should be 2.3, not 2.2.3. Revise. | We consider modelling to be part of the processing of data and thus should be a subchapter of 2.2 Data processing. So data modelling should be 2.2.3 |
| Figure 2: Please increase the size and/or the quality of the figure. It is very hard to observe it properly, even when zooming in on the pdf, let alone when printed… Use a better quality format (usually pdf format works great). Also, within the caption I don't think Fig. 2c should be written. (a) (b) (c) should be enough. | Good point. The figure has been both resized and improved in quality. |
| Table 2: Better % per decade, given the magnitude of the change per year. | Agree, this has been changed. |
| Line 149: Something odd in the structure of the sentence. ", of p values." What does it refer to? Same in Line 154, which is actually the same sentence… | Good point. The sentence was incomplete and should read „in terms of p values" this has been changed. |
| Line 156: "are" instead of "is". | Changed |
| Line 157: change to " SNCM = 4 (fully snow covered mountains)." | Changed |
| Line 160: full stop after observations. | Full stop added |
| Line 190: remove "fig" at the end. | Removed |
| Line 195: "were" instead of "are" at the end of the sentence. | „Are" changed to „were" |
| Line 224: remove "in" and "a". So "… a decrease in snow cover and snow mass across Iceland, …" | Removed. |

---

## Author Comment (AC2)

| Referee comment | Author response |
|---|---|
| General comments | |
| This manuscript presents an impressive multi-faceted analysis of the historical and (possible) future snow cover on Iceland based on in-situ observations, remotely sensed data, and climate simulations. The fact that much of the analysis, including the snow modeling, seems to have been carried out in the cloud was particularly novel to me and should be of great interest to the wider cryospheric community. | The authors are grateful to the referee for thorough and insightful comments. We have addressed all of the referees' general and specific comments and believe that this has resulted in substantial improvements to the paper. These include |
| | • Expansion of the introduction section e.g. to highlight more recent studies in the field. |
| Among the main results of this study, historical observations show an increasing trend in snow cover frequency (SCF) over Iceland during the last decades whereas projections from the simulations indicate that SCF will decrease substantially in the future especially under the more aggressive emissions scenario. This could suggest a threshold effect related to changes in precipitation phase under a warmer and wetter future climate. | • Revision and extension of the methods section e.g. clarify and elaborate of the modelling strategy and the description and processing of observational data. |
| | • Improved presentation and discussion of the results e.g. revision of all figures and the introduction of a new Figure 3 and discussion on the threshold effect mentioned by the referee. |
| | • Revisions to the discussion section for improved clarity. |
| While the manuscript was generally well written it is quite brief and parts of it felt a bit rushed. For example, the text could have benefited from more examples of recent related work on remote sensing and modeling for snow cover reanalysis (e.g. Alonso-González et al., 2021; Liu et al., 2021) and projections (Fiddes et al., 2022). | We agree with the referee that are many modelling strategies that can be applied to study snow cover. The objective of this study was to analyze trends in decadal average snow conditions over centennial time scales given of future climate scenarios derived by an ensemble of model projections. The modelling strategy used in this study was developed to optimally achieve that objective within the limits of the computational framework at our disposal. |
| The authors could also have taken advantage of their efficient conceptual snow model combined with downscaling (see Fiddes et al., 2022, and references therein) to conduct simulations at resolutions nearer to the hillslope scale (Fan et al., 2019) to try to better capture the spatial variability of the snowpack. Moreover, it did not seem as though the parameters in the SNOW-17 model were well calibrated using local information. This is a lost opportunity, as the MODIS data would have been a good candidate to calibrate many of the parameters (e.g. Fiddes et al., 2019; Alonso-González et al., | We agree that it would be a valuable effort to simulate Icelandic snow conditions at hillslope scale resolutions and that the MODIS data could be a good source to estimate some the SNOW-17 parameters at that resolution. This would most likely provide more accurate short-term simulations of the snowpack. However, as the objective of the study was to consider long term trends in average snow conditions given different emission scenarios from many models and several of the variables required for estimating the SNOW-17 parameters were not available at high spatial resolution (e.g. net SW) the model resolution was set at 0.2-degree. The parallel nature of the computing platform used in the study was also a limiting factor for model resolution when applied across centennial timescales for each member in a large ensemble of climate models. |

| | |
|---|---|
| 2021; Liu et al., 2021) such as the gauge under-catch factor and the range on the melt factor. This could have helped to constrain uncertainty in the SNOW-17 model and the forcing. | Furthermore, as the objective of the study was to study trends in long term snow conditions under a changing climate, the model was evaluated on its ability to simulate variability and average snow conditions on a decadal time scale. Figure 4 (now Figure 5) and the text describing it has been updated to highlight this point. |
| It was also odd that the GDDP forcing data was treated almost as a reanalysis dataset during the historical period rather than as (downscaled and bias-corrected) climate model output. In particular, this data is meant to represent the the correct climate on decadal timescales but not the day to day weather or even inter-annual variability. As such it is not fair to compare observations directly to SNOW-17 forced with this dataset. For such an exercise it would have made more sense to use actual reanalysis data, such as ERA5, to force the SNOW-17 model. Such simulations could be compared directly to the observations. Furthermore the climate (i.e. decadal moving averages) from the ERA5-driven simulations could be compared to the GDDP-driven simulations in the historical period (1950-2021) to gauge the performance of or bias correct the latter (Fiddes et al., 2022). | We agree that it would be an interesting research effort to apply the SNOW-17 model to reanalysis data such as ERA5 or GLDAS-2 and also at higher spatial resolution and that both should be addressed in future works. However, we feel that the extent of such a study would merit its own publication. We do note that reanalysis data from the GLDAS-2 dataset was used for parameter estimation, so weather data was used to adapt the model to the study area, this has been clarified in the text. |
| | We agree that the gap-filling product by Gunnarsson et al., 2019 provides an excellent analysis of plausible snow coverage. However, the approach taken in this study was to use the underlying research data as published without further manipulation such as e.g. interpolation of missing data. This has been clarified in the text. |
| I also had some concerns about the way the MODIS data was processed, particularly that no gap-filling was conducted before calculating SCF, despite previous work on such methods by some of the authors (Gunnarsson et al., 2019). This should at least be justified in the text. | Please see responses to the specific comments below. |
| As such, I suggest that this manuscript should undergo major revisions. Nonetheless, I would like to commend the authors for this valuable work that fits well within the scope of The Cryosphere and I encourage them to address these general comments and the specific comments below. | |

| Specific comments | |
|---|---|
| L9: Change "remote sensing observations" to "remotely sensed observations" (replace all). | This correction has been made throughout the manuscript |
| L10: Use either "General Circulation Model" or "Global Climate Model" | „General Circulation Model" is now used throughout the manuscript |

| | |
|---|---|
| (preferably the former) but not "Global Circulation Model". | |
| L11: One immediately wonders why the CMIP6 version of this dataset was not used. This is not a criticism per se, but the existence of a newer version of the NEX-GDDP dataset should at least be mentioned somewhere in the manuscript. | Good point. A mention of this has been added in the manuscript. |
| L13: Representative Concentration Pathway 4.5 (RCP4.5) indicates a scenario where the radiative forcing will be 4.5 Wm−2 at the end of the century. So please use the ".", it's RCP4.5 not RCP45. The same holds for RCP8.5. | Correct. „." Have been added to all mentions of RCP scenarios in the text. |
| • L13: Change "Snow17" to "SNOW-17" in line with Anderson's naming convention (replace all). | This correction has been made throughout the manuscript |
| L20: Suggest changing "climate" to "micro-climate" also change "significantly" to "strongly" since significant has a specific statistical meaning in your manuscript. | Good points, both corrections have been made. |
| L21: Change "correlate to changes in" to "are highly correlated with changes in". | This has been changed. |
| L30: Change "at least mid" to "at least the middle of the". | This has been changed |
| • L33: Suggest changing "duration of snow cover" to "snow cover duration" which is perhaps more widely used (e.g. Notarnicola, 2020). | We agree, this change has been made |
| L34: Best in what sense? A more qualified statement would be to say that a good balance of spatial and temporal resolution as well as temporal coverage is a valid reason for using MODIS. You could also consider citing a paper (e.g. Aalstad et al., 2020, and references therein) that evaluates satellite-based snow-covered area products from MODIS and give an idea of what uncertainty you would expect at the pixel scale. | We agree. This has been revised in the text (L40-41) |
| L40: Change "and is" to "which is". | Changed |
| • L41: Change "earths" to "Earth's". | Changed |
| • Fix typo "thatasc". | Changed |
| • L45: This could be a good place to mention other studies that have performed detailed future snow cover projections under different scenarios (e.g. Fiddes et al., 2022, and references therein). | Good point. The introduction has been restructured and extended to include a discussion of other studies of future snow cover projections. |
| L47: Change "Snow Covered Area" to "Snow-Covered Area". | This has been changed |

| | |
|---|---|
| • L49: It would be natural to mention that SNOW-17 is a conceptual model built around the degree day (also known as the temperature index) approach. Currently this is not mentioned anywhere in the manuscript. Since it is a degree day model, it is not only more efficient but it also has less requirements in terms of forcing data than full energy balance models. On the other hand, given that the entire energy balance is lumped into a single term, one would think that calibrating the degree day factor is critical for the model. | Good points. The end of the paragraph in question has been revised to illustrate these points.

We agree that calibration is beneficial for the optimal performance of any model in simulating the short-term response of a system to its environment. Here, the model parameters are a function of its environment which is expected to change over the long term, therefore, the initially calibrated model would be rendered biased over longer timescales, see e.g. Melsen and Guse, 2021. Therefore, the model parameters were estimated using best practices from globally available and comparable environmental datasets. |
| L51: Change "and it" to ". It". | Changed |
| L55: Change "calculated" to "estimated" since this is an approximation. | This has been changed |
| Section 2.1.2: The formulation in this section should be improved. A reader that is unfamiliar with these MODIS snow cover products could think that 'NDSI_Snow_Cover' is meant to measure Fractional SnowCovered Area (FSCA) directly. This is not the case, instead the valid pixels in the 'NDSI_Snow_Cover' field merely contains the NDSI value (scaled by 100, see e.g. Riggs and Hall (2020)) for pixels that could possibly contain snow (positive NDSI) or a value of 0 for pixels that probably do not contain snow (NDSI≤ 0) and have passed various screens (not deemed to be cloudy etc. . . ). This is readily verified by comparing it to the 'NDSI' field which will have equal values once negative values are set to zero and missing (cloudy etc...) pixels are masked out. The NDSI is of course related to the FSCA of a pixel, but they are not identical and converting NDSI to FSCA usually involves some form of transformation. In particular, it is often the case that even pixels with an intermediate positive NDSI value (say equal to 0.6) can be fully snow-covered (FSCA=1). It would be worth making this clear to the reader and discussing the commonly used linear relationship between the two (Salomonson and Appel, 2006; Fiddes et al., 2019; Alonso-González et al., 2021), along with some uncertainty estimates form the literature (Aalstad et al., 2020, | Good points. Section 2.1.2 has been restructured and revised to include the clarification between NDSI and FSCA and a discussion of commonly used transformations there between. |

| | |
|---|---|
| and references therein). Indeed an advantage with the C6 (versus C5) MODIS snow cover products is that users can customize the NDSI-FSCA relationship for their own use case (see the MODIS Snow Products Collection 6 User Guide user guide). Work with Sentinel-2 (Gascoin et al., 2020) has shown that other types of NDSI-FSCA relationships (e.g. sigmoid) can also perform well | |
| L73: How much did glacier outlines change during the MODIS era (2001-2021)? One would assume that this effect is small, but make it explicit that you assume these outlines to be constant in time for this period. | The changes to the glacier outlines are observed in our results, shown in Figure 3a (Now Figure 4a).

Glacier outlines were not used for modelling or analysis as the land surface was assumed to be constant. The reference to this dataset in the data section was inappropriate as it is only used for illustration in Figure 3 (Now Figure 4). |
| • L78: It would be nice to have at least a rough idea of what areal scales local snow cover and surrounding mountain snow cover represent? Moreover, the term "snow cover" is very general (see the NSIDC glossary) and a bit vague in this context in that it can implicitly refer to many different snow variables. You clarify your use of the term a bit later (L82) as being some kind of a snow cover classification (snow/patchy/no-snow) that lies between binary snow cover (snow/no-snow) and FSCA. I would still recommend calling it something more specific than "snow cover" such as "snow cover status" or similar | Good points.

Section 2.1.1. has been revised to describe the scale the measurements of SNC and SNCM are designed to represent.

The text was revised throughout to use the term „snow cover status" when referring to the SNC and SNCM snow cover observations. |
| L91: This sentence seems unnecessarily convoluted. Based on your classification system, an equivalent but more concise way to define "snow-covered ground" would be with the inequality SNC> 0 (or equivalently SNC≥ 2 since there is no class SNC= 1) and SNCM> 0 for local and surrounding mountain snow cover, respectively. | Agree and good point. This sentence has been revised and the classification system changed as per the referee suggestion. |
| L99: This is implicitly assuming that NDSI> 0 corresponds to FSCA> 0. While this is often true, in the sense that low but positive NDSI in a pixel can be due to a patchy snow cover, you may have some false positive 'snow-covered' pixels as a result of this, since some non-snow surfaces may also have a low but positive | We agree that assuming NDSI > 0 = FSCA > 0 holds the potential to include false negatives. We recognize the potential of an NDSI threshold value to exclude some valid observations.

In this study we made the approach to analyze the wealth of data available without further data manipulation to the data as it is published by the |

| | |
|---|---|
| NDSI. Did you look at the sensitivity of your results to this threshold, for example by trying a threshold of NDSI> 0.1 instead which may be less sensitive to such false positives. Moreover, you could quantify the false positive rate by for example comparing your MODIS-based snow cover status classification to that from the station measurements. This could help to calibrate the NDSI threshold. Perhaps it turns out that 0 is good choice of threshold threshold, but testing this can help to strengthen your analysis and the resulting conclusions. | original producers of the dataset, which has already undergone their internal QA.

Gunnarsson et al 2019 performed a validation of MODIS snow cover in Iceland using manned observations and higher spatial resolution remote sensing data (Landsat/Sentinel 2). Overall, a good agreement was found between the daily combined MODIS Terra∕Aqua data set and the validation data sets from Landsat 7/8, Sentinel 2 and in situ observations in Iceland.

To account for outliers in the MODIS snow cover data the Sens slope method was applied for trend analysis on the MODIS data as it is less sensitive to outliers and has shown good performance in snow cover trend analysis with the presence of outliers (Eythorsson et al., 2019). The text in section 2.2.4 has been extended to clarify this point. |
| L102: By excluding December and January from the analysis your SCF may end up incurring a negative bias (i.e. be underestimated). In particular, the months of December and January often have a higher SCF than many other months in the year. Since this is likely to be the case, your results may (at least on average spatially) end up being closer to the true annual SCF if you gap-filled your observations using either simple interpolation techniques or more sophisticated algorithms. More generally, applying gap-filling could also make your results more robust to a potentially uneven temporal distribution of cloudiness throughout the year. Given the large overlap in authorship, I found it surprising that you did not use the gap-filling method of Gunnarsson et al. (2019) which has already been applied successfully over your domain. | In this study we made the approach to analyze the wealth of data available in the area without performing any manipulation to the data from the original producers of the source datasets.

We acknowledge that systematic phenomena such as the polar night may introduce a negative interannual bias but recognize that the interpolation across large periods would also be a source of uncertainty in the SCF estimate.

As the study objective was to study and project long term trends in snow conditions under different climate scenarios and not investigate or develop methods for interpolating observational gaps to derive more plausible gridded snow cover datasets, we opted for using the unmanipulated data. This has been clarified in the text in Section 2.2.4 |
| To properly test the performance of SNOW-17 given weather (rather than climate) forcing I would strongly encourage the authors to also run SNOW-17 for the historical period (1950-2021) with forcing from the ERA5 reanalysis. Note that this product is also available on Google Earth Engine, so it should be relatively straightforward to extend your | We agree that testing the performance of the SNOW-17 model given weather forcing data such as e.g. those in the ERA5 or GLDAS-2 reanalysis products, both of which are available in the GEE data catalog would be interesting.

However, as the intent of this study was to investigate future snow conditions in Iceland given |

| | |
|---|---|
| analysis. This would be a fairer model simulation to compare with in-situ observations, which (disregarding scale-mismatches) experience (roughly) the same weather as in ERA5 rather than the weather simulated by GDDP which aims only to produce the correct climate (i.e. weather statistics). This ERA5 simulation would also a reference at roughly the same spatial scale as your existing GDDP-forced simulations that you could use to validate the climatic evolution (i.e. at the decadal timescale) of the snowpack in the historical period. | different climate scenarios, we opted to gauge the model performance in replicating the long-term average behavior Icelandic snow conditions.

Additionally, we note that the ERA5 daily aggregates published in the GEE data catalog extend from 1979 to the present and not from 1950, thus not covering as much of the historical period. (see https://developers.google.com/earth-engine/datasets/catalog/ECMWF_ERA5_DAILY#description) |
| L117: Change "of trend" to "of trends". | This has been changed |
| L119: It could be more instructive to cite studies that have applied similar trend analysis methods for snow cover such as Yılmaz et al. (2019) and Notarnicola (2020). | Good point. The citations have been revised. |
| • L120: This statement is misleading. When doing null hypothesis significance testing you are not testing the alternative hypothesis. You are not even really testing the null hypothesis. For more about the caveats of significance testing and the error of the transposed conditional see Ambaum (2010). To be more specific, I would urge the authors to explain what their p-values actually quantify. | We agree that this sentence was unclear. Section 2.2.4 was revised, the statement removed, and an explanation of the trend test p-values was added. |
| Figure 2: Please improve the resolution and visibility of this figure. For example, text should not be visibly pixelated and it should be easy to differentiate markers without zooming excessively | The size, resolution and visibility of the Figure 2 has been improved. |
| Suggest changing "ensemble average" to "ensemble mean" which is perhaps more commonly used. | We agree, this has been changed throughout the manuscript. |
| • L126-130: This is essentialy a repetition of the figure caption. Please shorten considerably, this should only briefly describe the results that the figure shows. | All figure and table captions have been revised for brevity and clarity. |
| L141 Change "trended upward" to "a positive trend". | Changed |
| L142 According to your earlier description the inequality describing full and patchy snow cover should be SNC≥ 2 not SNC> 2. | Correct, this has been corrected. |
| L146 Instead of speculating about what the increase in precipitation could lead to in terms of snow accumulation, you could analyze the changes in snowfall and | Good point. We have added a new analysis, presented in the new Figure 3 and text thereof to illustrate the effect of increased precipitation and |

| | |
|---|---|
| rainfall rates to a first order by applying simple air temperature-based thresholds to delineate precipitation phase. | temperature on the snow/rain temperature given a partition threshold. |
| Table 2: Please use × for multiplication not ∗. Also fix "of p-values Statiscially significant. . . " in the caption. Moreover, I would recommend setting all the text in the table to normal font apart from significant p-values which can be in bold. | Good points. All of these changes have been made. |
| Figure 3: Please center the colormap around 0. | Figure 3 (Now Figure 4) has been revised and the colormap centered around 0. |
| L176: Once more you are just repeating the caption. Please shorten considerably and avoid redundant text as much as possible. | All captions of tables and figures have been revised. |
| Figure 4: To make the long-term weather observations and the GDDP-driven climate simulations of SCF more comparable it would be natural to instead plot the decadal moving mean (serving as a low pass filter) of the observations. The corresponding moving standard deviation would then help to visualize the observed internal climate variability. In that way, you would be more fairly comparing observed climate to the simulated climate. In particular, as previously alluded to, GDDP is not a reanalysis so we can not expect it to reproduce the correct weather | Good point. Figure 4 (now Figure 5) has been updated to show the decadal moving average of the observations and the text describing it has been revised and expanded. The revisions highlight the fact that the objective of the study was to study long term climate impacts on Icelandic snow conditions. |
| • L190: Remove "fig". | This sentence has been revised |
| L192: Why aren't the observed and simulated trends compared for an overlapping period, such as 1950-2021? | The referee seems to have misunderstood.

L192 refers to the MODIS observations which are only available since 2001. Figure 4 (Now Figure 5) shows the IMO observations over the period compared to simulations for the period 1950-2021 and the MODIS observations for the period 2001-2021

The entire text around Figure 4 (Now Figure 5) has been revised for clarity on this and other points. |
| L204: What does "increase in snow cover" mean here? Are you referring to frequency, duration, depth, SWE? Please be more specific. | Good point. This sentence has been revised to refer specifically to SCF and 1st April SWE. |
| L205: This is not shown clearly in Figure 2, although maybe it will be easier to see when the Figure is sharpened. I would also recommend to add a panel to Figure 2 containing a scatter plot that compares | Figure 2 has been redone for better visibility. |

| | |
|---|---|
| the MODIS SCF to in-situ SCF observations for both local and mountain snow cover. | |
| L210: Please make it clear here that although the simulated SCF magnitude is comparable to the observations, the trend is in the opposite direction. | This has been clarified at this point and in the revised discussion Figure 4 (Now Figure 5). |
| L213: It's of course not straightforwards to directly compare SWE to snow depth given variations in snow density. Moreover, Figure 2 shows annual mean observed snow depth whereas Figure 4 shows simulated April 1 st SWE. Please clarify what you are comparing here. While the unrepresentative (mainly) low-lying station locations can account for some of the differences, you could perhaps make the simulations more comparable to the observations by excluding higher elevation grid cells | We agree that this comparison was unclear. This paragraph has been revised as well as the discussion of these results in text around Figures 4 (now 5) and 2. |
| L218: Once more "increase in snow cover" is vague, please explain what is meant by this | Agree, this has been changed to „increase in SCF" |
| L224: Change "predictions which forecast" to "projections of ". | Changed |
| L229: Change "parameters" to "variables". | Changed |
| L230: Although the acronym GHG is well-known, this is the first time it is used in this manuscript. Consider writing it out in full or just change "GHG emissions" to "emission scenarios". | Good point. This has been changed to „emission scenarios" |
| Code/Data availability: Please provide references (with DOIs) or link to all the datasets used. | Section 7 has been revised and extended. A table has been added with links to all underlying data used. |

---

## Referee Report (RR2)

**2$^{\text{nd}}$ round review of Eythorsson et al. (2022; doi:10.5194/egusphere-2022-590)**

**General comments**

In their revised manuscript and response the authors have successfully addressed the vast majority of my comments and concerns from the first round. As such, I only have a few technical corrections and very minor suggestions outlined below that I am confident the authors will address relatively easily. I would like to congratulate the authors on this work that would be a welcome contribution to *The Cryosphere*.

**Specific comments**

- L46: I was surprised to see the introduction of the term *"regional circulation model"* since it is completely new to me and seems to be uncommon in the literature. I would thus suggest that the authors switch this to *"regional climate model"*. If you are worried about consistency you can also switch *"general circulation model"* to the (roughly) synonymous *"global climate model"*.

- L85: The word *"parameter"* gives the impression of something that is more static than the dynamic properties that together describe the seasonal snowpack, such as snow depth, SWE, SNC, and SNCM. I would thus encourage the authors to instead refer to these as *"variables"* throughout the text. This would also be in line with the GCOS use of the term essential climate variables (ECVs) among which many snow variables feature.

- Figure 1: I do not recommend representing the number of stations as bar plots which (in my opinion) are rarely a good idea. The same information would be more clearly conveyed as line plots. This would also make the snow depth line plot more visible.

- L101: Be more specific here and change *"infrared"* to *"shortwave infrared"* since snow is still quite reflective in the near infrared. Consider also adding a reference to Dozier (1989) who was (to the best of my knowledge) the first to suggest the NDSI (without calling it that).

- L102 Change *"indicates"* to *"often indicates"*, otherwise you are overlooking false positives (i.e. NDSI$> 0$ when FSCA=0) which, despite being less frequent than true positives in cold regions, do occur.

- L108 For the advantages of spectral unmixing with higher resolution sensors I would recommend a reference to Cortés et al. (2014) as well.

- L118: Although it is perhaps implicitly obvious, consider adding that the CMIP6 version could be considered in future studies.

- L141: Change *"is temporal"* to *"is a temporal"*. Delete the second sentence here (*"It limits..."*) since there are many other applications of this data that are limited by polar night. If you want to keep it, say *"such as"* or provide a few more examples.

- L147: Thanks for clarifying the model resolution. As I understand it you are focusing on modeling the seasonal snowpack, yet nowhere does it state that you masked out glaciers in the observations (both MODIS and SNCM). I suspect this masking is done somehow, otherwise you would have a representativeness error (see e.g. Janjić et al., 2018) in your observations, but this needs to be stated somewhere in the manuscript.

- L151: This suggests that the SNOW-17 algorithm was coded in GEE by Anderson (2006), which is of course not the case. Instead, you could just change this to *"The SNOW-17 algorithm (Anderson, 2006)..."*

- L155: Change *"parameters must"* to *"parameters that must"*.

- L159: Change *"as followed as it incorporates"* to *"were followed as they incorporate"*.

- L162: Clarify what $"10 \times 10$ *DEM"* means in this context. I guess you mean 10 m resolution?

- As a side comment on Table 1, it is laudable that you seek to justify your choice of parameters with references and ancillary datasets. Nonetheless, this exercise is somewhat misleading as these are effective parameters that compensate for the physical processes that are missing in a degree day model. As such, they do not really have universal values that can be extracted from look-up-tables (or similar) based on ancillary data. Instead, their "optimal" values will likely vary considerably based on the forcing data used and other details of the experimental setup such as the spatial resolution of the model. My issue is that this presentation may (inadvertently) camouflage the uncertainty that exists in these parameters rather than embracing it and attempting to calibrate them with the data (e.g. MODIS snow cover) that you have at hand. This becomes especially apparent when one of two references in Table 1 is from a conference abstract, was there no subsequent peer-reviewed publication by these authors that goes into more detail on their methods? In summary, although I understand that calibration is beyond the scope of this work, the uncertainty introduced by the particular choice of parameters should at least be mentioned somewhere in the manuscript.

- L174: This sentence is problematic. I don't see why published datasets would be less uncertain than derived datasets that seek to make improvements. Please change the formulation here. It is fine to say that for simplicity and to stay within the scope of your study you used these datasets without further manipulation.

- L180: To me at least, this is not an explanation of what your p-values measure. Instead, you have merely passed the problem along to the term *"95% confidence level"*. Their are plenty of definitions floating around, see e.g. Ambaum (2010) or Benjamin et al. (2018), and I would urge the authors' to adapt the wording in these to their particular tests. The reason I recommend this is that null hypothesis significance testing plays a central part in your results (the term significant is mentioned 4 times in Section 4 alone), so I believe it is important that you properly define what is meant by this term.

- L182: Change *"ensemble average"* to *"ensemble mean"*.

- L270: Please use $\times$ not $*$ for multiplication when using scientific notation, i.e. $p = 1.54 \times 10^{-5}$.

- L277: Since you are using *"significant"* in a very specific technical sense elsewhere, I would recommend changing this (different) usage of the word to *"considerable"* or similar.

- L290: Unsure how helpful it is to introduce the acronym CESM2 which you then never use elsewhere.

- Table 3 (and elsewhere): I am not sure why you are citing Hall et al. (2006) since you are using the V006 MODIS snow cover product not the V005 product. The correct citation is surely Hall et al. (2016) throughout the manuscript. Also consider following the suggested citation for Hall et al. (2016) on the product webpage at NSIDC from which yours deviates slightly.

Kind regards,
Kristoffer Aalstad

**References**

Ambaum, M. H. P.: Significance Tests in Climate Science, Journal of Climate, https://doi.org/10.1175/2010JCLI3746.1, 2010.

Benjamin, D. et al.: Redefine statistical significance, Nature Human Behaviour, https://doi.org/10.1038/s41562-017-0189-z, 2018.

Cortés, G., Girotto, M., and Margulis, S.: Analysis of sub-pixel snow and ice extent over the extratropical Andes using spectral unmixing of historical Landsat imagery, Remote Sensing of Environment, https://doi.org/10.1016/j.rse.2013.10.023, 2014.

Dozier, J.: Spectral Signature of Alpine Snow Cover from the Landsat Thematic Mapper, Remote Sensing of Environment, https://doi.org/10.1016/0034-4257(89)90101-6, 1989.

Janjić, T. et al.: On the representation error in data assimilation, Quarterly Journal of the Royal Meteorological Society, https://doi.org/10.1002/qj.3130, 2018.

---

## Author Response (AR2)

The authors are grateful to the referees for the through feedback and constructive comments provided in the review process. All comments have been addressed, as specified in the table below, and have resulted in significant improvements to the manuscript.

| Referee#1 Comments | Author response |
|---|---|
| Lines 67-71: Good to add this small paragraph as summary of the novelty. I suggest removing lines 70 and 71, so finishing the paragraph at " […] trusted snow model." | Agree. The text in lines 70-71 was redundant and has been removed. |
| Line 58: I think it should be snowpack | This has been changed |
| Line 139 "is A temporal limitation" | This has been corrected. |
| Line 140: Full stop at the end. | Full stop added |
| Line 151. "model parameters THAT must be specified" | Correct. "that has been added" |
| Line 152. I guess it's topography, not topology. | Correct. Topology changed to topography |
| Line 155. I think it should be snowmelt (Also in line 267 and 269). | Agree. Snow melt changed to snowmelt throughout |
| Line 156: was followed. | This has not been changed. This sentence refers to the recommendations of Andersson, 2006 and should be in plural. |
| Line 158: were, not was. | This has been changed. |
| Line 159: 10x10 meters? | Correct. This has been changed. |
| Table 1. (and anywhere else if I missed it) Units must be written exponentially as per The Cryosphere submission guidelines. | Correct. Units have been revised to be written exponentially throughout the manuscript. |
| Line 166: "… number OF times A model grid cell … " | This has been changed. |
| Figure 2b and 2c: I asked the authors to provide a plot of number of observational stations in time and the mean snow depth of those stations over time. This is now in Figure 1 right panel. It can be seen that before 1960 there were only around 5 stations measuring snow depth over Iceland. This dramatically increases after 1960. Those few stations before 1960 seem to have a shallower snow depth than the stations in the period after 1960 (judging by the dark line in the plot). If a trend is computed over the yearly average of all stations in Iceland, this can give an inaccurate trend. Perhaps those 5 stations were located at coastal Iceland (lower snow depth), and the stations from 1960 are located more inland (higher snow depth). The trends from 1930-2021 would therefore be artificial if it is computed over the average snow depth of all stations for each year. The correct thing would be to do the | Good points. The trends and their estimated significance of the trends illustrated in Figure 2b and 2c presented in Table 2 have been updated based on the historical period 1950-2021 as you are correct that the number of measurements is much lower and less representative prior (mostly costal locations). We have also made the same period changes to Figures 2b and 2c so as to a) show only collectively representative snow data as suggested by the referee and the number of IMO stations reporting snow data significantly increased in the 1950's (as illustrated in Figure 1) and b) the period starting in 1950 corresponds both to the NEX-NDDP data shown in Figure 2a) and the period of the SNOW-17 simulations performed in the study. |

| average trend, instead of the trend of the average. That means, calculate a trend for each of the IMO stations, then show the average trend. This should be done for the results in Figure 2b and 2c. Alternatively, trends could be computed only from 1960, with higher confidence that the distribution of stations is not changing significantly. | We have made a note of the collective representativeness of data prior to 1950 in the text describing Figure 2 in Lines 202-204 |
|---|---|
| Line 208: Why temperature in K? I think it should be in Celsius. | Agree. This has been changed. |
| Figure 3 caption: specify which model simulations are these. | Good point. The caption to Figure 3 has been revised to include specifications about model simulations. |
| Figure 4: Even though the authors have addressed my comment here, something still looks odd. The colour scale of SCF change in 4a goes from +100% to -100% (is this supposed to be per decade?). Judging from the colours, some areas in Iceland are quite close to +100% change over the period 2001-2021. This seems too much since values in table 2 show that the MODIS trends are in the order of 2-4% per decade, which sounds more sensible. I believe either the labels or some calculation is wrong. | Table 2 shows the average SCF trend across the country for the same period which smooths out the variation observed in Figure 4. So, although some pixels may have large changes in SCF over the period the average change across the country will be much lower.

Also, the color scale in Figure 4 colors the relative SCF over the whole period 2001-2021 within in each pixel, not the ΔSCF. This has been clarified in the text

Lastly, The 100% values on the color scale were chosen to encompass all observations and for simplicity. We note that 100% is uncommon and the highest values observed in Figure 4 are related to areas where recent surface changes are responsible for the change in SCF, e.g. at the termini of the outlet glaciers where the receding glacier has caused SCF to drop considerably from the historical 100%. This has been clarified in the text. |
| Line 260. I suggest to rephrase: "… , their trend is opposite. The observational data show an increasing SCF trend while the simulations show a decreasing trend over the historical period 1950- 2020." | Good point. Thanks! This sentence has been rephrased |
| Line 269: "leading to A countrywide" | This has been changed |

| Referee#2 Comments | Author response |
|---|---|
| L46: I was surprised to see the introduction of the term "regional circulation model" since it is completely new to me and seems to be | The text has been revised to refer to Regional Climate Models |

| | |
|---|---|
| uncommon in the literature. I would thus suggest that the authors switch this to "regional climate model". If you are worried about consistency you can also switch "general circulation model" to the (roughly) synonymous "global climate model". | |
| L85: The word "parameter" gives the impression of something that is more static than the dynamic properties that together describe the seasonal snowpack, such as snow depth, SWE, SNC, and SNCM. I would thus encourage the authors to instead refer to these as "variables" throughout the text. This would also be in line with the GCOS use of the term essential climate variables (ECVs) among which many snow variables feature. | Good point. The text has been revised to refer to "variables" |
| Figure 1: I do not recommend representing the number of stations as bar plots which (in my opinion) are rarely a good idea. The same information would be more clearly conveyed as line plots. This would also make the snow depth line plot more visible. | Good point. The plot has been changed from bars to lines. |
| L101: Be more specific here and change "infrared" to "shortwave infrared" since snow is still quite reflective in the near infrared. Consider also adding a reference to Dozier (1989) who was (to the best of my knowledge) the first to suggest the NDSI (without calling it that). | Good points. Thanks. The text has been revised to refer to "short wave infrared" and the reference to Dozier has been added. |
| L102 Change "indicates" to "often indicates", otherwise you are overlooking false positives (i.e. NDSI> 0 when FSCA=0) which, despite being less frequent than true positives in cold regions, do occur. | This has been changed. |
| L108 For the advantages of spectral unmixing with higher resolution sensors I would recommend a reference to Cortés et al. (2014) as well | Good point. The reference has been added |
| L118: Although it is perhaps implicitly obvious, consider adding that the CMIP6 version could be considered in future studies. | This has been added |
| L141: Change "is temporal" to "is a temporal". Delete the second sentence here ("It limits. . . ") since there are many other applications of this data that are limited by polar night. If you want to keep it, say "such as" or provide a few more examples. | These changes have been made. |
| L147: Thanks for clarifying the model resolution. As I understand it you are focusing on modeling the seasonal snowpack, yet nowhere does it state | The glaciers were not masked out in the observations nor the simulations. Figure 4 shows the RGI glacier outlines for reference but the |

| | |
|---|---|
| that you masked out glaciers in the observations (both MODIS and SNCM). I suspect this masking is done somehow, otherwise you would have a representativeness error (see e.g. Janjić et al., 2018) in your observations, but this needs to be stated somewhere in the manuscript. | MODIS data did include the glaciers (SCF = 100). The SNCM data is specifically collected by the IMO for mountaintops observed from mostly lowland areas some of whom do not loose all snow every year. However, as most of the glaciated areas are in the interior highland and not directly visible from most IMO sites this would not cause a significant error. |
| L151: This suggests that the SNOW-17 algorithm was coded in GEE by Anderson (2006), which is of course not the case. Instead, you could just change this to "The SNOW-17 algorithm (Anderson, 2006). . . " | Good point. This has been changed and the reference to Anderson 2006 moved for clarification. |
| L155: Change "parameters must" to "parameters that must". | This has been changed |
| • L159: Change "as followed as it incorporates" to "were followed as they incorporate". | This has been changed |
| • L162: Clarify what "10 × 10 DEM" means in this context. I guess you mean 10 m resolution? | Correct. This has been clarified. |
| As a side comment on Table 1, it is laudable that you seek to justify your choice of parameters with references and ancillary datasets. Nonetheless, this exercise is somewhat misleading as these are effective parameters that compensate for the physical processes that are missing in a degree day model. As such, they do not really have universal values that can be extracted from look-up-tables (or similar) based on ancillary data. Instead, their "optimal" values will likely vary considerably based on the forcing data used and other details of the experimental setup such as the spatial resolution of the model. My issue is that this presentation may (inadvertently) camouflage the uncertainty that exists in these parameters rather than embracing it and attempting to calibrate them with the data (e.g. MODIS snow cover) that you have at hand. This becomes especially apparent when one of two references in Table 1 is from a conference abstract, was there no subsequent peer-reviewed publication by these authors that goes into more detail on their methods? In summary, although I understand that calibration is beyond the scope of this work, the uncertainty introduced by the particular choice of parameters should at least be mentioned somewhere in the manuscript. | Good point. This has been clarified in the text in lines 159-162 . |

| | |
|---|---|
| L174: This sentence is problematic. I don't see why published datasets would be less uncertain than derived datasets that seek to make improvements. Please change the formulation here. It is fine to say that for simplicity and to stay within the scope of your study you used these datasets without further manipulation. | Agree. This sentence has been revised. L174. |
| L180: To me at least, this is not an explanation of what your p-values measure. Instead, you have merely passed the problem along to the term "95% confidence level". Their are plenty of definitions floating around, see e.g. Ambaum (2010) or Benjamin et al. (2018), and I would urge the authors' to adapt the wording in these to their particular tests. The reason I recommend this is that null hypothesis significance testing plays a central part in your results (the term significant is mentioned 4 times in Section 4 alone), so I believe it is important that you properly define what is meant by this term. | Good points. Thanks. This sentence has been revised in line 182-184 for clarification. |
| L182: Change "ensemble average" to "ensemble mean". | This has been changed. |
| L270: Please use × not ∗ for multiplication when using scientific notation, i.e. $p = 1.54 \times 10^{-5}$ . | This has been changed. |
| L277: Since you are using "significant" in a very specific technical sense elsewhere, I would recommend changing this (different) usage of the word to "considerable" or similar. | This has been changed. |
| L290: Unsure how helpful it is to introduce the acronym CESM2 which you then never use elsewhere. | Good point. The acronym has been removed. |
| Table 3 (and elsewhere): I am not sure why you are citing Hall et al. (2006) since you are using the V006 MODIS snow cover product not the V005 product. The correct citation is surely Hall et al. (2016) throughout the manuscript. Also consider following the suggested citation for Hall et al. (2016) on the product webpage at NSIDC from which yours deviates slightly. | Good points. This has been changed. |